# Stochastic Linear Bandits with Parameter Noise

Daniel Ezer [1]    Alon Peled-Cohen [1 2]    Yishay Mansour [1 2]

## Abstract

We study the stochastic linear bandits with parameter noise model, in which the reward of action $a$ is $a^\top \theta$ where $\theta$ is sampled i.i.d. We show a regret upper bound of $\widetilde{O}(\sqrt{dT \log(K/\delta)\sigma_{\max}^2})$ for a horizon $T$, general action set of size $K$ of dimension $d$, and where $\sigma_{\max}^2$ is the maximal variance of the reward for any action. We further provide a lower bound of $\widetilde{\Omega}(d\sqrt{T\sigma_{\max}^2})$ which is tight (up to logarithmic factors) whenever $\log(K) \approx d$. For more specific action sets, $\ell_p$ unit balls with $p \leq 2$ and dual norm $q$, we show that the minimax regret is $\widetilde{\Theta}(\sqrt{dT\sigma_q^2})$, where $\sigma_q^2$ is a variance-dependent quantity that is always at most $4$. This is in contrast to the minimax regret attainable for such sets in the classic additive noise model, where the regret is of order $d\sqrt{T}$. Surprisingly, we show that this optimal (up to logarithmic factors) regret bound is attainable using a very simple explore-exploit algorithm.

## 1. Introduction

Linear bandits have many practical applications, such as personalized news recommendations (Li et al., 2010), medical treatments (Lansdell et al., 2019), online advertisement (Tian, 2025) and stock trading (Ji et al., 2024), to name a few. In addition, there is substantial theoretical work on linear bandits, includingAbbasi-yadkori et al. (2011); Lattimore & Szepesvári (2020); Slivkins (2024); Auer (2003).

Stochastic linear bandits is an online learning problem in which at each timestep $t$, the learner chooses an action $a_t \in \mathcal{A} \subseteq \mathbb{R}^d$, and receives a stochastic reward $X_t$. We assume that the expectation of $X_t$ is a linear function of the action $a_t$, and the learner's goal is to maximize their cumulative reward. Linear bandits generalize the classical multi-armed bandit (MAB) setting from finite action sets, in which actions correspond to standard basis vectors, to large or even infinite action spaces, by assuming that the reward function is linear.

The most widely studied stochastic model is the *additive noise* model. In this model, the stochastic reward is given by $X_t = a_t^\top \theta^\star + \eta_t$, where $\theta^\star$ is an unknown reward vector, and $\eta_t$ is a zero-mean sub-Gaussian random variable.

An alternative learning model is the *adversarial* model, in which the reward is given by $X_t = a_t^\top \theta_t$, where the reward vectors $\theta_t$ are chosen by an adversary. Intuitively, we expect the stochastic model to be easier than the adversarial model, due to the worst case nature of the adversary. However, when the action set is the $\ell_2$ unit ball, this is surprisingly not the case. Bubeck et al. (2012) showed that in the adversarial model, for a horizon $T$, one can get a regret bound of $\widetilde{O}(\sqrt{dT})$, while Rusmevichientong & Tsitsiklis (2010) and Lattimore & Szepesvári (2020, Theorem 24.2) proved a lower bound of $\Omega(d\sqrt{T})$ in the additive noise model. This surprising "contradiction" is the topic of a blog post by the name *the curious case of the unit ball* (Lattimore & Szepesvári, 2016), and has a dedicated chapter in the seminal bandit algorithms book (Lattimore & Szepesvári, 2020, Chapter 29).

To build intuition for the seemingly contradictory upper and lower bounds and how they are reconciled, consider a simple and natural reduction between the two models. Let $\widetilde{\mathcal{ALG}}$ be a bandit algorithm with regret of $\widetilde{O}(\sqrt{dT})$ in the adversarial model on action set $\mathcal{A}$, which is the $\ell_2$ unit ball. We want to build a bandit algorithm $\mathcal{ALG}$ that has the same regret bound in the additive noise model. The natural reduction is to define an action set $\mathcal{B} = \mathcal{A} \times \{1\}$ and let $\widetilde{\mathcal{ALG}}$ play on $\mathcal{B}$. Since the regret is guaranteed for any arbitrary sequence of reward vectors, we can choose $\theta_t = (\theta^\star, \eta_t)$. Thus, when $\widetilde{\mathcal{ALG}}$ selects an action $b_t = (a_t, 1) \in \mathcal{B}$, $\mathcal{ALG}$ plays $a_t$ and gets reward $X_t = b_t^\top \theta_t = a_t^\top \theta^\star + \eta_t$, which is exactly the reward in the additive noise model. However, it is important to note that our reduction changed the action set - while $\mathcal{ALG}$ plays on $\mathcal{A}$, $\widetilde{\mathcal{ALG}}$ plays on $\mathcal{B}$. Since regret bounds in linear bandits are extremely sensitive to the choice of the action set, even though $\widetilde{\mathcal{ALG}}$ has a regret bound of $\widetilde{O}(\sqrt{dT})$ on $\mathcal{A}$, it has no guarantee for $\mathcal{B}$. Moreover, Shamir (2014) even proved a lower bound for $\mathcal{B}$ of

---

[1]Tel Aviv University [2]Google Research. Correspondence to: Daniel Ezer <danielezer@mail.tau.ac.il>.

*Proceedings of the 43$^{rd}$ International Conference on Machine Learning*, Seoul, South Korea. PMLR 306, 2026. Copyright 2026 by the author(s).

$\widetilde{\Omega}(d\sqrt{T})$, showing that this reduction cannot achieve the desired regret.

In this work, we study the stochastic linear bandits with *parameter noise* model (as coined by Lattimore & Szepesvári, 2020, Chapter 29), which has received little attention from the research community. We assume that the reward for an action $a_t$ is given by $X_t = a_t^\top \theta_t$, where $\theta_t$ is a reward vector sampled from some fixed unknown distribution $\nu$, with expectation $\theta^\star$ and covariance matrix $\Sigma$. This model can be motivated by online advertising, if we consider $\theta_t$ as a feature vector that represents a user sampled from the population, and our goal is to find an advertising strategy that works well in expectation across all users.

Unlike the additive noise model, the parameter noise model is trivially reducible to the adversarial model. Thus, any regret upper bound for the adversarial model also holds for the parameter noise model, making it more aligned with the intuition about stochastic and adversarial models.

Importantly, while the two stochastic models are different, there is a simple reduction between them, showing that, unsurprisingly, the additive noise model is harder than the parameter noise model. Namely, the parameter noise model can be written as $X_t = a_t^\top \theta^\star + \eta_t$, where $\eta_t = a_t^\top \theta_t - a_t^\top \theta^\star$. We can see that indeed $\mathbb{E}[\eta_t] = 0$, and since we assumed the rewards are bounded, we also have that $|\eta_t| \leq 2$, and so $\eta_t$ is 4-sub-Gaussian. This implies that any regret bound for the additive noise model applies to the parameter noise model as well.

However, the two stochastic models are not equivalent, and have two key differences between them. First, in the parameter noise model, not only the expectation of the reward is linear, but also its realization. This makes the reward more structured compared to the additive noise model, in which the realization is a misspecified linear function. Second, in the parameter noise model, the learner can influence the variance of the reward. That is, when the learner chooses an action $a_t$, it can be shown that $\mathrm{Var}(X_t) = a_t^\top \Sigma a_t$. This differs from the additive noise model, in which $\mathrm{Var}(X_t) = \mathbb{E}[\eta_t^2]$, which is completely determined by the environment.

Building on this observation, we study the parameter noise model and provide insights to the achievable regret bounds for a general action set, and for specific action sets. While the additive noise model might be harder than the adversarial model, we show that the parameter noise model is never harder, and prove that in some cases, one can get much better regret bounds compared to the adversarial model.

**Our Contributions**

We present two algorithms - one for a general finite action set, and one for $\ell_p$ unit balls with $p \leq 2$. The former, shown in Algorithm 2, is a successive elimination algorithm that can also be used for a general infinite set, by using a covering argument (see, e.g., Dani et al., 2008). The latter, a very simple explore-exploit type algorithm, presented in Algorithm 3, achieves better regret bounds for some action sets and is specifically designed for $\ell_p$ unit balls. We prove variance dependent regret bounds for both algorithms, that are minimax optimal.

More specifically, denoting $\sigma_{\max}^2$ as the maximal variance of an action in $\mathcal{A}$, we have the following main results, summarized in Table 1:

(i) For a general finite action set of size $K$, Algorithm 2 incurs regret of $\widetilde{O}(d^2 + \sqrt{dT \log(K/\delta) M_\sigma})$ where $M_\sigma \leq \sigma_{\max}^2$. A more refined bound can be found in Theorem 3.1. For a general action set, this bound is optimal up to logarithmic factors, as shown by Theorem 4.4.

(ii) When the action set is the $\ell_p$ unit ball for $p \in (1, 2]$ and dual norm $q \geq 2$:
  - If the covariance matrix is known, Algorithm 3 has a regret bound of $\widetilde{O}(d + \sqrt{dT\sigma_q^2})$, where $\sigma_q^2$ is a variance dependent quantity to be defined in Section 2. This is accompanied by a lower bound that shows that this result is optimal up to logarithmic factors (Theorems 3.7 and 4.3).
  - If the covariance matrix is unknown, Algorithm 3 incurs regret of $\widetilde{O}(\sqrt{dT\sigma_q^2} + d^{2/3+2/3q}T^{1/3})$, which is optimal up to the logarithmic factors and the additive $d^{2/3+2/3q}T^{1/3}$ term (Theorem 3.9).

**1.1. Previous Work**

The stochastic linear bandits with parameter noise model has seen little attention in the community. Mostly, research has focused on the additive noise model and the adversarial model. Since the parameter noise model can be seen as a relaxation of the adversarial model, we also review the relevant results in the adversarial setting, in which the reward vectors $\theta_t$ are chosen by an adversary.

**Stochastic linear bandits with additive noise.** In the additive noise model, we assume that the reward of action $a_t$ satisfies $X_t = a_t^\top \theta^\star + \eta_t$, where $\theta^\star$ is some unknown reward vector and $\eta_t$ is a zero-mean 1-Sub-Gaussian noise. There has been ample research in this model, both for a finite action set (Auer, 2003; Chu et al., 2011; Chapter 22 in Lattimore & Szepesvári, 2020) and for infinite action sets (Dani et al., 2008; Rusmevichientong & Tsitsiklis, 2010; Abbasi-yadkori et al., 2011). In the case of a finite action set of size $K$, an optimal design exploration based algorithm appearing in Lattimore & Szepesvári (2020, Chapter 22) achieved a regret bound of $\widetilde{O}(\sqrt{dT \log(K)})$. For the case of an infinite action set, Dani et al. (2008) showed

*Table 1.* Summary of our results

| Algorithm | Regret | Action Set | Covariance Matrix | Theorem |
|---|---|---|---|---|
| VASE (Algorithm 2) | $\widetilde{O}(d^2 + \sqrt{dT \log(K/\delta)\sigma_{\max}^2})$ | General finite set | Known & Unknown | Theorem 3.1 |
| VALEE (Algorithm 3) | $\widetilde{O}(d + \sqrt{dT\sigma_q^2})$ | $\ell_p$ unit ball, $p \leq 2$ | Known | Theorem 3.7 |
| VALEE (Algorithm 3) | $\widetilde{O}(dT^{1/3} + \sqrt{dT\sigma_q^2})$ | | Unknown | Theorem 3.9 |
| Lower bound | $\widetilde{\Omega}(d\sqrt{T\sigma_{\max}^2})$ | $\ell_p$ unit ball, $p > 2$ | Known | Theorem 4.4 |
| Lower bound | $\widetilde{\Omega}(\sqrt{dT\sigma_q^2})$ | $\ell_p$ unit ball, $p \leq 2$ | | Theorem 4.1 |

that a regret bound of $\widetilde{O}(d\sqrt{T})$ is achievable for any action set, however the algorithm might not be efficient. Rusmevichientong & Tsitsiklis (2010) showed that a simple and efficient explore-then-commit algorithm has a regret bound of $d\sqrt{T}$ for strongly convex action sets (such as the $\ell_2$ unit ball), under some additional assumptions. Specifically, they assumed that when playing on the $\ell_2$ unit ball, $\|\theta^\star\|_2$ is bounded away from 0, which generally might not be the case. These results matched the lower bounds in Lattimore & Szepesvári (2020, Chapter 24) for the case where $\mathcal{A}$ is the hypercube or the $\ell_2$ unit ball.

**Bayesian Linear Bandits.** The Bayesian linear bandits model is similar to the additive noise model, with the added assumption that $\theta^\star$ is sampled from some prior distribution $\mathcal{D}$. Russo & Van Roy (2014) proved that with a finite action set of size $K$, one can bound the Bayesian risk with $\widetilde{O}(\sqrt{dT\mathcal{H}(\mathcal{D})})$, where $\mathcal{H}$ is the Shannon entropy. Since $\mathcal{H}(\mathcal{D}) \leq \log(K)$, this can be bounded by $\widetilde{O}(\sqrt{dT \log(K)})$, which matches the lower bound of Dani et al. (2007) for a general finite action set. For an infinite action set, using a covering argument, the risk of Russo & Van Roy (2014) becomes $\widetilde{O}(d\sqrt{T})$, which complies with the bound for the additive noise model. This also matches the Bayesian risk lower bound from Rusmevichientong & Tsitsiklis (2010).

**Adversarial linear bandits.** In adversarial linear bandits, we assume the reward of action $a_t$ is given by $X_t = a_t^\top \theta_t$, where $\theta_t$ is chosen by an adversary. In this model, the adversary is allowed to choose the reward vectors arbitrarily, while in the parameter noise model the reward vectors are generated according to a fixed distribution. Thus, the parameter noise model can be seen as a relaxation of the adversarial model. Bubeck et al. (2017) showed regret bounds of $\widetilde{O}(\sqrt{dT})$ for $\ell_p$ unit balls where $p \leq 2$, and $\widetilde{O}(d\sqrt{T})$ for $p > 2$. These results are also accompanied by matching lower bounds (up to logarithmic factors). Namely, this shows that, unlike in the additive noise model, when the action set is an $\ell_p$ unit ball for $p \leq 2$, it is possible to get better regret bounds compared to the general case. Shamir (2014) showed that these results are surprisingly brittle, by proving that a slightly modified action set, e.g. a capped $\ell_2$ unit ball, can change the achievable regret from $\sqrt{dT}$ to $d\sqrt{T}$.

Bubeck et al. (2012) focused on the finite armed case, and proved a regret bound of $\widetilde{O}(\sqrt{dT \log(K)})$. Since the parameter noise model is trivially reducible to the adversarial model, the upper bounds hold in our model as well. While the lower bounds don't generally apply to the easier parameter noise model, Shamir (2014); Bubeck et al. (2017) both use a stochastic adversary that aligns with our model in their construction. Hence, the lower bounds in both papers hold for the parameter noise model as well.

**Variance dependent regret bounds.** In the additive noise model, there is a line of work focusing on regret bounds that depend on the variances of the arms. Indeed, these works formulate the intuition that smaller reward variance leads to lower regret bounds. To our knowledge, the first to consider leveraging variance information are Zhou et al. (2021). They presented an algorithm which has a regret bound of $\widetilde{O}(\sqrt{dT} + d\sqrt{\sum_{t=1}^{T} \sigma_t^2})$, where $\sigma_t^2$ is the known variance of the arm chosen by the algorithm at time $t$. Zhou & Gu (2022) presented a refined regret bound of $\widetilde{O}(d + d\sqrt{\sum_{t=1}^{T} \sigma_t^2})$. This result is tight and matches the lower bound of He & Gu (2025) up to logarithmic factors.

Subsequently, Zhang et al. (2021) removed the known variance assumption, and achieved a regret bound of $\widetilde{O}(\sqrt{d^{4.5} \sum_{t=1}^{T} \sigma_t^2 + d^5})$. Next, Kim et al. (2023) improved the dimension dependence to $\widetilde{O}(d^2 + d^{1.5}\sqrt{\sum_{t=1}^{T} \sigma_t^2})$. More recently Zhao et al. (2023) presented the SAVE algorithm, and proved $\widetilde{O}(d + d\sqrt{\sum_{t=1}^{T} \sigma_t^2})$ regret without knowledge of the variances, matching the lower bound of He & Gu (2025).

It is important to note that all the regret bounds mentioned in this section are algorithmic-dependent, in the sense that the regret bound is a random variable that depends on the arms chosen by the algorithm. Building on the reduction between the additive and parameter noise models in Section 1, we can write $\eta_t = a_t^\top(\theta_t - \theta^\star)$, and hence $\sigma_t^2 = \text{Var}(\eta_t) = a_t^\top \Sigma a_t$. Thus, using the SAVE algorithm (Zhao et al., 2023) in the parameter noise model gives a regret bound of $\widetilde{O}(d\sqrt{\sum_{t=1}^{T} a_t^\top \Sigma a_t})$. Since the algorithm can play arms with large variance, this can amount to

$\widetilde{O}(d\sqrt{T\max_{a\in\mathcal{A}}a^\top\Sigma a})$, even on the $\ell_2$ unit ball, which is worse than our bound.

## 2. Preliminaries

**Learning model**. We consider *stochastic linear bandits with parameter noise* over an action set $\mathcal{A}\subseteq\mathbb{R}^d$. The interaction protocol is as follows, with some fixed distribution $\nu$, for $t=1,\dots,T$:

- The learner chooses an action $a_t\in\mathcal{A}$.
- The environment samples a random vector $\theta_t\sim\nu$ independently of previous timesteps.
- The learner gets reward $X_t=a_t^\top\theta_t$.

Throughout this work, we assume that for any $\theta\sim\nu$ and $a\in\mathcal{A}$ we have that $|a^\top\theta|\le 1$ almost surely. We denote the expectation and covariance matrix of the reward distribution $\nu$ by $\theta^\star$ and $\Sigma$, respectively.

The learner's goal is to minimize their regret, which is defined as the difference between the reward of the best fixed action and the learner's cumulative reward. Formally, we define the (pseudo) regret of a bandit algorithm $\mathcal{ALG}$ with time horizon $T$ as follows:

$$\mathcal{R}_T(\mathcal{ALG})=\sum_{t=1}^{T}(a^\star-a_t)^\top\theta^\star,$$

where $a^\star\in\arg\max_{a\in\mathcal{A}}a^\top\theta^\star$.

**Notations**. Throughout this paper, we denote $\sigma_q^2=\left(\sum_{i=1}^{d}\Sigma_{ii}^{q/2}\right)^{2/q}$, where $\Sigma_{ii}$ is the $i$-th diagonal entry of $\Sigma$. For any $a\in\mathcal{A}$, we denote $\sigma^2(a)=a^\top\Sigma a$ and $\sigma_{\max}^2=\max_{a\in\mathcal{A}}\sigma^2(a)$. We also define the sub-optimality gap by $\Delta_a=(a^\star-a)^\top\theta^\star$.

**G-Optimal design**. A policy, or design, $\pi$ is a function $\pi\in\mathcal{A}\mapsto[0,1]$ such that $\sum_{a\in\mathcal{A}}\pi(a)=1$. Let $g(\pi)=\max_{a\in\mathcal{A}}\|a\|_{V(\pi)^{-1}}$, where $V(\pi)=\sum_{a\in\mathcal{A}}\pi(a)aa^\top$. A G-optimal design (Lattimore & Szepesvári, 2020, Chapter 21), is a policy $\pi^\star$ that minimizes $g(\pi)$. Kiefer & Wolfowitz (1960) proved that $g(\pi^\star)=d$, and that there exists a minimizer $\pi^\star$ with $\mathrm{Supp}(\pi^\star)\le d(d+1)/2$. Moreover, Frank & Wolfe (1956); Fedorov (1972) proved the existence of an approximately optimal design $\pi$ for which $\mathrm{Supp}(\pi)=O(d\log\log d)$ and $g(\pi)\le 2d$. G-optimal designs are used to bound the variance of the least squares estimator in linear experiment design, and were proven crucial for attaining optimal regret bounds in linear bandits (see, e.g., Lattimore & Szepesvári, 2020; Bubeck et al., 2012). Geometrically, optimal designs have been shown to be equivalent to the minimum volume enclosing ellipsoid (MVEE) of a convex set, where the support of the design is the contact points with the ellipsoid's boundary (Silvey & Sibson, 1972; Lat-

timore & Szepesvári, 2020).

**Stopping Rule algorithm**. The stopping rule algorithm (Dagum et al., 1995), which we henceforth reference as $\mathcal{SR}$, is a Las Vegas algorithm for estimating the expectation of a random variable. Given samples from $X\in[0,1]$ with expectation $\mu>0$, $\mathcal{SR}$ gives a simple and efficient way to get an estimator $\hat{\mu}$ that is a high probability multiplicative estimate for $\mu$. Concretely, given an approximation error $\varepsilon$ and confidence level $\delta$, $\mathcal{SR}$ repeatedly samples from $X$ and yields an estimator $\hat{\mu}$ such that with probability $1-\delta$ we have that $|\hat{\mu}-\mu|\le\varepsilon\mu$. Additionally, with probability $1-\delta$, the algorithm stops after $O(\log(1/\delta)/\varepsilon^2\mu)$ steps.

**Explore-Exploit.** One of our algorithms (Algorithm 3) is inspired by the simple explore-exploit algorithm (Slivkins, 2024, Algorithm 1.1). In the original explore-exploit algorithm, each arm is played $M$ times for some predefined fixed $M$, and the algorithm then exploits the arm with the best empirical average reward. This extremely simple and easy to implement algorithm can be shown to achieve suboptimal regret of $K^{1/3}T^{2/3}$ where $K$ is the number of arms. In fact, Slivkins (2024, Theorem 2.11) shows that this bound is tight for any algorithm that fixes $M$ ahead of time. Surprising and unlike the case of MAB, we show that our explore-exploit Algorithm 3 is optimal.

## 3. Algorithms and Proof Sketches

We present two algorithms - an optimal design exploration based algorithm for a general finite action set, and a simple explore-exploit type algorithm for $\ell_p$ unit balls. In this section we briefly explain the ideas behind the two algorithms, state our results and provide proof sketches. The complete proofs can be seen in Appendices A and B, respectively.

### 3.1. VASE

VASE (Algorithm 2) is designed for finite action sets, and is intended to achieve a regret bound that depends on the variance of the arms. When the variance is small, this can yield a significant improvement compared to the general regret bound of $\widetilde{O}(\sqrt{dT\log(K)})$ which doesn't make use of variances.

The algorithm is inspired by Lattimore & Szepesvári (2020, Algorithm 12), and is a successive elimination-based algorithm for finite action sets, which leverages variance information for shorter exploration. In each phase $\ell$, we first find a G-optimal design $\pi_\ell$ and estimate the variance of each action in the support of $\pi_\ell$. We use the estimated variance twice - once for reducing the number of samples we need from each action, and once as weights for the linear regression estimator $\hat{\theta}^{(\ell)}$. Using the estimator, we eliminate suboptimal actions from $\mathcal{A}$ in each phase, and continue with a subset $\mathcal{A}_\ell\subseteq\mathcal{A}$ of potentially optimal arms.

---

**Algorithm 1** Variance estimation

---

**Require:** action $a$, threshold $\tau$, confidence $\bar{\bar{\delta}}$ and error $\bar{\varepsilon}$
1: Define $Z_s(a) = \max\{\frac{1}{4}(X_s - X_{s+1})^2, \frac{\tau}{2}\}$, where $X_s$ and $X_{s+1}$ are the rewards from playing $a$ twice.
2: Run $\mathcal{SR}(\bar{\varepsilon}, \bar{\bar{\delta}})$ on samples from $Z_s(a)$ and get estimator $\hat{Z}(a)$.

---

VASE is presented in Algorithm 2, and its regret bound is stated in the following theorem, proved in Appendix A:

**Theorem 3.1.** *With probability* $1 - 3\delta$, *the regret of Algorithm 2 with a finite action set of size $K$ is bounded by:* $\widetilde{O}\left(d^2 + \sqrt{dT\log(K/\delta)M_\sigma}\right)$, *where* $M_\sigma = \min\left\{\max_{a\in\mathcal{A}} \sigma^2(a), \max_{a\in\mathcal{A}}\|a\|_2^2\mathrm{tr}(\Sigma)\right\}$.

*Remark* 3.2. We can use the Frank-Wolfe algorithm (Frank & Wolfe, 1956) to obtain a design that 2-approximates $g(\pi^\star)$, with a smaller support of size $d\log\log(d)$. This will replace the $d^2$ term in the regret with $d\log(\log(d)) + d\log(K)$, which is a better dependency on $d$.

Theorem 3.1 shows the dependency between the achievable regret and the geometry of the action and reward sets. Specifically, the second term in the minimum defining $M_\sigma$ shows that the regret of Algorithm 2 depends on the geometry of the action set through $\max_{a\in\mathcal{A}}\|a\|_2^2$, and on the reward set via $\mathrm{tr}(\Sigma)$. Moreover, when the action set has large $\ell_2$ norm, which holds for e.g. $\ell_p$ unit balls with $p > 2$, the regret is controlled by the maximal variance. Thus, we have the following corollary:

**Corollary 3.3.** *With probability* $1 - 3\delta$, *the regret of Algorithm 2 on $\mathcal{A}$ which is a discretization of the $\ell_p$ unit ball for $p > 2$ is bounded by:* $\widetilde{O}\left(d^2 + d\sqrt{T\log(1/\delta)\sigma_{\max}^2}\right)$.

*Remark* 3.4. The regret bound in Corollary 3.3 matches our lower bound in Theorem 4.4, up to logarithmic factors. This shows that Algorithm 2 is able to achieve the optimal regret bound when run on an $\ell_p$ unit ball with $p > 2$.

### 3.1.1. ANALYSIS AND PROOF SKETCH

We now provide a short proof sketch that our algorithm achieves the regret bound in Theorem 3.1. The complete proof can be seen in Appendix A. We assume throughout that $\mathcal{A}_\ell$ spans $\mathbb{R}^d$. This assumption is not limiting, since similarly to Lattimore et al. (2020, Remark 5.2), we can simply work in the subspace spanned by $\mathcal{A}_\ell$.

In the regret analysis of the original algorithm (Lattimore & Szepesvári, 2020, Algorithm 12), it can be shown that $d/\varepsilon_\ell^2$ exploration steps in each phase are sufficient for attaining $\varepsilon_\ell$ error. Here, we show that our variance-aware implementation requires $d\sum_{a\in\mathcal{A}_\ell}\sigma^2(a)\pi_\ell(a)/\varepsilon_\ell^2$ steps, which is always fewer than $d/\varepsilon_\ell^2$, since $\pi_\ell$ is a design and $\sigma^2(a) \leq 1$.

---

**Algorithm 2** VASE - **V**ariance **A**ware Optimal Design **S**uccessive **E**limination

---

1: $\mathcal{A}_1 \leftarrow \mathcal{A}, \ell \leftarrow 1, t \leftarrow 1, t_1 \leftarrow 1$.
2: **while** $t \leq T$ **do**
3:    $\varepsilon_\ell \leftarrow 2^{-\ell}, \delta_\ell \leftarrow \frac{\delta}{K\ell(\ell+1)}, \gamma_\ell = \frac{2\delta}{\ell(\ell+1)d(d+1)}$.
4:    Find G-optimal design $\pi_\ell \in \mathcal{A}_\ell \mapsto [0,1]$
5:    **Estimate the variance:**
6:    **for** each arm $a \in \mathrm{Supp}(\pi_\ell)$ **do**
7:       Run Algorithm 1 with $(a, \varepsilon_\ell, \gamma_\ell, 1/2)$.
8:       Let $\hat{\sigma}_\ell^2(a)$ be its output and $T_\ell^{\mathcal{SR}}(a)$ its number of steps.
9:    **end for**
10:   $\forall a \in \mathrm{Supp}(\pi_\ell)$, set:

$$T_\ell(a) \leftarrow \left\lceil \frac{49d}{\varepsilon_\ell^2} \cdot \log(1/\delta_\ell) \cdot \hat{\sigma}_\ell^2(a)\pi_\ell(a) \right\rceil.$$

     and play each action $T_\ell(a)$ times.
11:   Calculate weighted least squares estimator:

$$\hat{\theta}^{(\ell)} = V_\ell^{-1} \sum_{s=t_\ell}^{t_\ell+T_\ell} \hat{\sigma}_\ell^2(a_s)^{-1} a_s X_s,$$

     where:   $V_\ell = \sum_{s=t_\ell}^{t_\ell+T_\ell} \hat{\sigma}_\ell^2(a_s)^{-1} a_s a_s^\top$  and $T_\ell = \sum_{a\in\mathcal{A}_\ell} T_\ell(a)$
12:   Eliminate arms:

$$\mathcal{A}_{\ell+1} = \left\{ a \in \mathcal{A}_\ell : \max_{b\in\mathcal{A}_\ell}(b-a)^\top\hat{\theta}^{(\ell)} \leq 2\varepsilon_\ell \right\}$$

13:   $t \leftarrow t + T_\ell, \ell \leftarrow \ell + 1, t_\ell \leftarrow t$.
14: **end while**

---

When the variances are small, this can significantly shorten the exploration.

Formally, we first prove that our variance estimation is good using the following lemma:

**Lemma 3.5.** *With probability* $1 - \delta$, *the following hold for all* $\ell \in \mathbb{N}^+$ *and* $a \in \mathrm{Supp}(\pi_\ell)$ *jointly:* $\max\{\varepsilon_\ell, \sigma^2(a)\}/4 \leq \hat{\sigma}_\ell^2(a) \leq 3\sigma^2(a)/4 + 3\varepsilon_\ell/4$.

Then, we prove that our estimator has an error of $\varepsilon_\ell$, as stated in the following lemma:

**Lemma 3.6.** *With probability* $1 - \delta$, *for all* $\ell \geq 2$ *and* $a \in \mathcal{A}_\ell$ *jointly:* $|a^\top(\hat{\theta}^{(\ell)} - \theta^\star)| \leq \varepsilon_\ell$.

In order to prove Lemma 3.6, we use Freedman's inequality (Freedman, 1975) to bound the random variable $a^\top(\hat{\theta}^{(\ell)} - \theta^\star)$ for some action $a \in \mathcal{A}_\ell$ with high probability. The crux of the proof that helps us achieve variance dependent regret is the weighted least squares. Our choice of weights makes sure that the covariance of the least squares estimator doesn't change, even though we play each ac-

tion less times. This yields the same concentration error of $\|a\|_{V_\ell^{-1}}$ as in Lattimore & Szepesvári (2020), while performing fewer exploration steps.

The next steps of the proof are showing that the optimal action is not eliminated (Lemma A.11) and that actions we don't eliminate are almost optimal (Lemma A.12). Using these lemmas, we bound the regret of each phase $\ell$ by $\sum_{a \in \mathcal{A}_\ell} \hat\sigma_\ell^2(a)/\varepsilon_\ell$. Summing over all $\ell$, we upper bound this with $\widetilde{O}\left(\sqrt{dT \sum_{\ell=1}^M \sum_{a \in \mathcal{A}_\ell} \pi_\ell(a)\sigma^2(a)}\right)$. The sum can in turn be bounded by $\log(T)M_\sigma$, which yields our regret bound.

### 3.2. VALEE

For the case in which $\mathcal{A}$ is the $\ell_2$ unit ball, we can use Algorithm 2 with a covering argument using a finite $\mathcal{A}$ of size $\Omega(\epsilon^{-d})$, which causes the regret in Theorem 3.1 to scale as $d\sqrt{T\mathrm{tr}(\Sigma)}$. By our lower bound in Theorem 4.1, we know that this is suboptimal. Thus, in this section we consider a different algorithm that is designed for infinite action sets that include the standard basis vectors, such as $\ell_p$ unit balls.

VALEE (Algorithm 3), is similar in spirit to the basic explore-exploit algorithm in classic MAB (Slivkins, 2024, Algorithm 1.1), in the sense that it separates the exploration and exploitation stages. However, our exploration is adaptive - it depends on the norm of the expected reward vector and its variance. This simple approach yields an efficient and easy-to-implement algorithm that can achieve optimal regret (up to logarithmic factors) when the action set is an $\ell_p$ unit ball for $p \in (1, 2]$ and the covariance matrix is known.

When the covariance matrix is known, VALEE operates in two phases - exploration and exploitation. The exploration phase plays only the standard basis vectors $e_i$, and uses the results to build an estimator for the expected reward vector $\theta^\star$. The estimation is done per coordinate using the median of means (MoM) method, in which we build $\kappa$ independent estimators and use their median as the final estimator. This phase continues until our estimation error is small enough compared to the number of steps and variance of the rewards.

Next, the exploitation phase commits to an action $\hat{a}$, as defined in Algorithm 3, and plays it until the time is up. The reason for choosing this action is that the true optimal action $a^\star$ is given by: $a_i^\star = \mathrm{sign}(\theta_i^\star)|\theta_i^\star|^{q-1}/\|\theta^\star\|_q^{q-1}$, and so $\hat{a}$ is exactly $a^\star$ with $\theta^\star$ replaced by $\hat\theta^{(j)}$. If we think of $\hat\theta^{(j)}$ as being a good estimator for $\theta^\star$, then $\hat{a}$ is also a good estimator for $a^\star$. We state this formally in Lemma 3.12.

When the covariance matrix is unknown, Algorithm 3 has an additional phase, preceding the exploration, for estimating the variance. In this phase, we use the $\mathcal{SR}$ algorithm to

---

**Algorithm 3** VALEE - **V**ariance **A**ware **L**inear **E**xplore **E**xploit

**Require:** Threshold $\tau$ and confidence $\delta$.
1: **if** The covariance matrix is unknown: **then**
2:     **Estimate the variance:**
3:     for all $i \in [d]$, run Algorithm 1 with $(e_i, \tau, \delta/d, 1/2)$ and get $\hat\sigma^2(e_i)$.
4:     Define $\hat\sigma_q^2 = \left(\sum_{i=1}^d (\hat\sigma^2(e_i))^{q/2}\right)^{2/q}$.
5: **end if**
6: **Explore:**
7: Set $\kappa \leftarrow \lceil 8\log(d\log(T)/\delta) \rceil$ and:

$$\alpha \leftarrow \left(\frac{d\kappa}{Tq\hat\sigma_q^2}\right)^{1/4}.$$

8: Initialize: $j \leftarrow 0$, $\|\hat\theta^{(j)}\|_q \leftarrow 0$, $\hat{N}_j \leftarrow 2$, $t_j \leftarrow 1$.
9: **while** $\hat{N}_j \geq \|\hat\theta^{(j)}\|_q$ **do**
10:     Set: $t_{j,\ell} \leftarrow t_j$, $\hat{N}_j \leftarrow \frac{1}{2}\hat{N}_{j-1}$, $\hat\varepsilon_j \leftarrow \alpha\sqrt{\hat{N}_j}$, $T_j^{exp}(e_i) \leftarrow \lceil 8/\hat\varepsilon_j^2 \rceil$, $T_j^{exp} \leftarrow \sum_{i=1}^d T_j^{exp}(e_i)$ and $V_j = \sum_{s=t_j}^{t_j+T_j^{exp}} a_s a_s^\top$.
11:     **while** $\ell \leq \kappa$ **do**
12:         Play $e_i$ for $T_j^{exp}(e_i)$ time steps, for $i = 1, \ldots, d$ and define:

$$\hat\theta_i^{(j,\ell)} = \frac{1}{T_j^{exp}(e_i)} \sum_{s=t_{j,\ell}}^{t_{j,\ell}+T_j^{exp}(e_i)} X_s.$$

13:         Update $t_{j,\ell} \leftarrow t_{j,\ell} + T_j^{exp}(e_i)$.
14:     **end while**
15:     Define estimate $\hat\theta^{(j)}$ where $\hat\theta_i^{(j)}$ is chosen by the median of means technique with $\kappa$ means.
16:     Update $t_j \leftarrow t_j + \kappa T_j^{exp}$
17: **end while**
18: **Exploit:**
19: Play $\hat{a}$ for all remaining timesteps, where:

$$\hat{a}_i = \frac{\mathrm{sign}(\hat\theta_i^{(j)})|\hat\theta_i^{(j)}|^{q-1}}{\|\hat\theta^{(j)}\|_q^{q-1}}.$$

---

estimate $\sigma^2(e_i)$ for all $i \in [d]$ and use these estimates for shortening the length of the exploration phase.

The regret bound of Algorithm 3 when the covariance matrix is known, is given by the following theorem, proved in Appendix B:

**Theorem 3.7.** *Assume that the covariance matrix $\Sigma$ is known. Then with probability $1 - 2\delta$, the regret of Algorithm 3 on the $\ell_p$ unit ball for $p \in (1, 2]$ and dual norm $q \geq 2$, is bounded by:* $\widetilde{O}\left(d + \sqrt{dTq\log(1/\delta)\sigma_q^2}\right)$.

*Remark* 3.8. This result matches our lower bound in Theorem 4.1 up to logarithmic factors.

When the covariance matrix is unknown, the regret is as stated in the following theorem, proved in Appendix B:

**Theorem 3.9.** *Assume that the covariance matrix $\Sigma$ is unknown. Then with probability $1 - 2\delta$ the regret of Algorithm 3 on the $\ell_p$ unit ball for $p \in (1, 2]$ and dual norm $q \geq 2$, is bounded by:*

$$\widetilde{O}\Big(d^{2/3+2/3q} \log^{2/3}(1/\delta)(Tq)^{1/3} + \sqrt{dTq \log(1/\delta)\sigma_q^2}\Big).$$

### 3.2.1. ANALYSIS AND PROOF SKETCH

To provide intuition to our algorithm, we focus on the $\ell_2$ unit ball. The main idea in our analysis is leveraging the parameter noise model to show that exploring each standard basis vector for $1/\varepsilon^2$ samples yields an estimation error of $\varepsilon\sqrt{\sigma^2(e_i)}$. Previous work in the additive noise model showed that the same number of samples yields an error of $\varepsilon$, which is always worse than our error bound. Formally, this is stated in the following lemma, proved for the case of $p \in (1, 2]$ in Lemma B.9:

**Lemma 3.10.** *With probability $1-\delta$, for all $i \in [d]$ and $j \leq \log(T)$ jointly, we have that: $e_i^\top(\hat{\theta}^{(j)} - \theta^\star) \leq \sqrt{\sigma^2(e_i)}\hat{\varepsilon}_j$.*

Equipped with this improved concentration bound, we leverage the geometry of the action set and Cauchy-Schwarz to show that we can get an $\ell_2$ estimation error of $\varepsilon\sqrt{\mathrm{tr}(\Sigma)}$, whereas in the additive noise the same argument would result in an error of $\varepsilon\sqrt{d}$. This is stated in the following lemma, proved for the case of $p \in (1, 2]$ in Lemma B.10:

**Lemma 3.11.** *With probability $1 - \delta$, for all $j \leq \log(T)$ jointly, we have that: $\|\hat{\theta}^{(j)} - \theta^\star\|_2 \leq \sqrt{\mathrm{tr}(\Sigma)}\hat{\varepsilon}_j$.*

In Lemma B.3 we show that $\mathrm{tr}(\Sigma) \leq 1$. Moreover, when the variances are small, $\mathrm{tr}(\Sigma)$ can be much smaller, leading to a significantly improved error bound.

Next, we prove that the $\ell_2$ error between $\hat{a}$ (the action that Algorithm 3 commits to) and the optimal action, is controlled by the approximation error of $\hat{\theta}^{(j)}$. Intuitively, this is true since the $\ell_2$ unit ball is curved and has a smooth surface, and so slightly changing the reward vector implies a slight change in the optimal action. This is in contrast to, e.g., the simplex, which represents MAB, that has a flat surface. More formally, we prove the following lemma, proved for the case of $p \in (1, 2]$ in Lemma B.13:

**Lemma 3.12.** *If Algorithm 3 reached the exploit stage after $L$ iterations, then with probability $1 - \delta$ we have:*

$$\|\hat{a} - a^\star\|_2 \leq \frac{3\sqrt{\mathrm{tr}(\Sigma)}\hat{\varepsilon}_L}{\|\theta^\star\|_2}.$$

We use Lemma 3.12 to upper bound the sub-optimality of $\hat{a}$, which determines the regret of the exploit phase. This is stated in the following lemma, proved for the case of $p \in (1, 2]$ in Lemma B.14:

**Lemma 3.13.** *If Algorithm 3 reached the exploit stage after $L$ iterations, then with probability $1 - \delta$, the suboptimality gap of $\hat{a}$ is bounded by:*

$$\Delta_{\hat{a}} \leq \frac{3\mathrm{tr}(\Sigma)\hat{\varepsilon}_L^2}{\|\theta^\star\|_2}.$$

This error bound improves the standard concentration analysis which shows that the same amount of samples yields $\varepsilon$ error. The reason for the improvement in our result is the linear structure of the reward function and the geometry of the action set, which has some curvature when $p = 2$.

Using Lemma 3.13, ignoring constants and logarithmic factors, a simple analysis shows that the regret is bounded by: $d\varepsilon^2/\|\theta^\star\|_2 + T\mathrm{tr}(\Sigma)\varepsilon^2/\|\theta^\star\|_2$. If we know $\|\theta^\star\|_2$ but don't know $\mathrm{tr}(\Sigma)$, choosing $\varepsilon \approx (\|\theta^\star\|_2/T)^{1/4}$ yields a regret bound of $(1 + \mathrm{tr}(\Sigma))\sqrt{dT} = O(\sqrt{dT})$.

However, assuming we know $\|\theta^\star\|_2$ is not realistic, and so the initial phase of Algorithm 3 is devoted to estimating the norm using a doubling argument. We start by "guessing" that the norm is 1, and perform the number of exploration steps required when this is true. Then, we build an estimator $\hat{\theta}^{(j)}$ and use its norm as an estimate for $\|\theta^\star\|_2$. We stop doubling when the estimator's norm is small compared to the number of exploration steps, and then continue to the exploitation stage. We show that we can approximate $\|\theta^\star\|_2$ in the following lemma, proved for the case of $p \in (1, 2]$ in Lemma B.11:

**Lemma 3.14.** *If the exploration stage of Algorithm 3 stops in iteration $L$, then with probability $1 - \delta$ we have that: $\|\theta^\star\|_2/4 \leq \hat{N}_L \leq 5\|\theta^\star\|_2/2$.*

Using this result, we can show that we can get a regret bound of $O(\sqrt{dT})$, even without knowledge of $\|\theta^\star\|_2$.

Thus far, we were able to achieve a regret bound matching that of the adversarial case. Next, we will leverage the variance information in order to improve this bound. Building up to our actual algorithm, we first assume we know $\mathrm{tr}(\Sigma)$, and so can choose $\varepsilon$ more subtly to depend on $\mathrm{tr}(\Sigma)$. Choosing $\varepsilon \approx (\|\theta^\star\|_2/T\mathrm{tr}(\Sigma))^{1/4}$ yields a variance dependent regret bound of $O(d + \sqrt{dT\mathrm{tr}(\Sigma)})$, which might be much smaller than $O(\sqrt{dT})$ when the variance is small. In fact, when the covariance matrix is known, this bound is optimal up to logarithmic factors, as we later show a matching lower bound (Theorem 4.1).

Finally, dropping the assumption that we know $\mathrm{tr}(\Sigma)$, we use a simple algorithm (Algorithm 1), to estimate it in order to tune $\varepsilon$. We show that we can get an estimation for

$\text{tr}(\Sigma)$, with an additive error that depends on a threshold $\tau \approx dT^{-1/3}$, stated precisely in Appendix B. For the case of $p = q = 2$, the lemma is as follows, and we prove it for the case of $p \in (1, 2]$ in Lemma B.7:

**Lemma 3.15.** *With probability* $1 - \delta$*, the following hold jointly:*

- *The variance estimation is good:* $\text{tr}(\Sigma)/4 \leq \hat{\sigma}_q^2 \leq 3d\tau/2 + 3\text{tr}(\Sigma)/2$.
- *The regret of the variance estimation is bounded by:* $180d \log(2d/\delta)/\tau$.

As stated in Lemma 3.15 and by the choice of $\tau$, the variance estimation phase incurs an additive regret penalty of $dT^{1/3}$. This bound is still always better than the standard $\sqrt{dT}$ regret bound, but when the variance is extremely small (smaller than $T^{-1/3}$), one might be able to get a better regret bound. We leave closing this gap to future work, as noted in Section 5.

## 4. Lower Bounds

In this section, we present our lower bounds for the parameter noise setting, which depend on the variance of the reward distribution. We state three lower bounds - one for the case in which the action set is the $\ell_p$ unit ball for $p \leq 2$, and two for the case where $p > 2$. We also provide short proof sketches for our bounds. The full proofs can be seen in Appendix C.

While Bubeck et al. (2017) show lower bounds of $\sqrt{dT}$ for $\ell_p$ unit balls with $p \in (1, 2]$ and $d\sqrt{T}$ for $p > 2$ in the adversarial setting, we present a more refined version that depends on $\sigma_q^2$ - a measure of the reward variance. If we plug in a distribution with constant $\sigma_q^2$ into our lower bound, we get lower bounds for the parameter noise model of $\sqrt{dT}$ and $d\sqrt{T}$ for $p \leq 2$ and $p > 2$, respectively. Since the parameter noise model is a relaxation of the adversarial model, this also implies the same lower bound for the adversarial setting. This is the exact proof methodology that Bubeck et al. (2017) chose, by designing a stochastic adversary that samples reward vectors from a fixed distribution with constant $\sigma_q^2$. Thus, our result, which depends on $\sigma_q^2$, can be seen as a generalization of their result to the case of a general covariance matrix.

Formally, our lower bounds are stated in the following theorems:

**Theorem 4.1.** *For any bandit algorithm* $\mathcal{ALG}$ *running on the* $\ell_p$ *unit ball for* $p \in (1, 2]$ *and dual norm* $q \geq 2$*,* $T \geq d$ *and any* $\sigma^2 > 0$*, there exists a reward distribution* $\mathcal{D}$ *with* $\left(\sum_{i=1}^{d} \Sigma_{ii}^{q/2}\right)^{2/q} = \sigma^2$ *such that* $\mathcal{R}_T(\mathcal{ALG}, D) = \widetilde{\Omega}\left(\sqrt{dT\sigma^2}\right)$*.*

*Remark* 4.2. Theorem 4.1 shows that our result from Theorem 3.7 is optimal, up to logarithmic factors.

**Theorem 4.3.** *For any bandit algorithm* $\mathcal{ALG}$ *running on the* $\ell_p$ *unit ball for* $p > 2$ *and dual norm* $q \in [1, 2)$*,* $T \geq d^{\frac{4}{2-q}}$ *and any* $\sigma^2 > 0$*, there exists a reward distribution* $\mathcal{D}$ *with* $\sigma_q^2 = \sigma^2$ *such that* $\mathcal{R}_T(\mathcal{ALG}, D) = \widetilde{\Omega}\left(d\sqrt{T\sigma^2}\right)$*.*

We note that Theorem 4.3 can also be stated as follows, in terms of the maximal variance:

**Theorem 4.4.** *For any bandit algorithm* $\mathcal{ALG}$ *running on the* $\ell_p$ *unit ball for* $p > 2$ *and dual norm* $q \in [1, 2)$*,* $T \geq d^{\frac{4}{2-q}}$ *and any* $\sigma^2 > 0$*, there exists a reward distribution* $\mathcal{D}$ *with* $\sigma_{\max}^2 = \sigma^2/2$ *such that* $\mathcal{R}_T(\mathcal{ALG}, D) = \widetilde{\Omega}\left(d\sqrt{T\sigma^2}\right)$*.*

Our construction and proof of Theorem 4.1 follow a similar method to that presented in Shamir (2014), by randomly sampling a vector $\xi \in \{-1, 1\}^d$, and choosing $\mathcal{D}$ to be a Gaussian distribution centered around $\varepsilon\xi$, for some small $\varepsilon$. Carefully choosing the covariance matrix as the scalar matrix with the largest possible variance in each direction, and setting $\varepsilon = d^{1/2-1/q}\sigma/\sqrt{T}$, yields our lower bound.

This shows that the regret of Algorithm 3, stated in Theorem 3.7, is optimal up to logarithmic factors, when the covariance matrix is known.

For Theorem 4.3, we adapt the proof technique and construction of Bubeck et al. (2017) to our model. We choose a Gaussian distribution $\mathcal{D}$ over $\mathbb{R}^{d+1}$, whose expectation is $(1, \varepsilon\xi)$, where $\xi$ is again sampled uniformly from $\{-1, 1\}^d$. The covariance matrix is chosen such that the variance of the first coordinate is roughly equal to that of the other coordinates combined. The intuition, as described by Bubeck et al. (2017), is that the learner must choose between exploration and exploitation rounds. In exploration rounds, the learner tries to learn the sign of $\xi$, whereas in exploitation rounds they place constant weight on the first coordinate.

Bubeck et al. (2017) then show that in order to achieve low regret, the learner must find the sign of $\xi$, and so exploitation is not sufficient. Since exploring, i.e. choosing a small value in the first coordinate, also incurs large regret, the learner must balance exploration and exploitation. Following their argument and carefully choosing $\varepsilon$ such that $\varepsilon \approx \sigma/\sqrt{T}$ yields our regret lower bound.

## 5. Discussion

In this work, we focused on the stochastic linear bandits with parameter noise model, and showed that for some action sets, we can get better regret bounds compared to the adversarial model.

We also presented novel lower bounds that generalize previous bounds to the case of a general covariance matrix of the reward distribution. This sheds more light on the opti-

mal achievable regret in the parameter noise model.

While our results for the $\ell_p$ unit ball with $p \leq 2$ are optimal (up to logarithmic factors) in the case of a known covariance matrix, we incur an additive $dT^{1/3}$ term when the covariance matrix is unknown. We believe this cannot be mitigated with an explore-exploit algorithm, and finding an algorithm which is optimal when the covariance matrix is unknown is left to future work.

Additionally, for $\ell_p$ unit balls with $p > 2$, we present the lower bound of $d\sqrt{T\sigma_q^2}$, and a natural and interesting open question is to find an algorithm whose regret bound matches this bound.

## Acknowledgments

This project has received funding from the European Research Council (ERC) under the European Union's Horizon 2020 research and innovation program (grant agreement No. 882396), by the Israel Science Foundation (1357/24 and 2250/22), the Yandex Initiative for Machine Learning at Tel Aviv University and a grant from the Tel Aviv University Center for AI and Data Science (TAD).

## Impact Statement

This paper presents work whose goal is to advance the field of Machine Learning. There are many potential societal consequences of our work, none which we feel must be specifically highlighted here.

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

# A. Regret Bound Proof for VASE

In this section, we prove the regret bound for Algorithm 2, as stated in Theorem 3.1. We start by setting some notation and proving some helpful lemmas.

## A.1. Notations

Throughout our proof, we use the following notation:

$$\hat{B}_\ell = \sum_{a \in \mathcal{A}_\ell} \hat{\sigma}_\ell^2(a)\pi_\ell(a), \qquad B_\ell = \sum_{a \in \mathcal{A}_\ell} \sigma^2(a)\pi_\ell(a)$$

We also denote enote by $M$ the number of phases performed by the algorithm, and denote $T_\ell^{\mathcal{SR}} = \sum_{a \in \mathcal{A}_\ell} T_\ell^{\mathcal{SR}}(a)$.

## A.2. Proof

We start by proving a few lemmas.

**Lemma A.1.** $B_\ell = \mathrm{tr}(\Sigma V(\pi_\ell))$

*Proof.* By the linearity and cyclic invariance of the trace, we have:

$$B_\ell = \sum_{a \in \mathcal{A}_\ell} \sigma^2(a)\pi_\ell(a) = \sum_{a \in \mathcal{A}_\ell} a^\top \Sigma a \cdot \pi_\ell(a) = \sum_{a \in \mathcal{A}_\ell} \mathrm{tr}\left(\Sigma \pi_\ell(a) a a^\top\right)$$
$$= \mathrm{tr}\left(\Sigma \sum_{a \in \mathcal{A}_\ell} \pi_\ell(a) a a^\top\right) = \mathrm{tr}(\Sigma V(\pi_\ell))$$

Where the first equality is by the definition of $\sigma^2(a)$. $\qquad\square$

**Lemma A.2.** *for all $\ell$ we have that:*
$$\max_{a \in \mathcal{A}_\ell} \|a\|_{V_\ell^{-1}} \leq \frac{\varepsilon_\ell}{7\sqrt{\log(1/\delta_\ell)}}$$

*Proof.* By our choice of $T_\ell$ and since $g(\pi) \leq d$ (Kiefer & Wolfowitz, 1960), we have:

$$\max_{a \in \mathcal{A}_\ell} \|a\|_{V_\ell^{-1}} \leq \frac{\varepsilon_\ell}{7\sqrt{d\log(1/\delta_\ell)}} \max_{a \in \mathcal{A}_\ell} \|a\|_{V(\pi_\ell)^{-1}} \leq \frac{\varepsilon_\ell}{7\sqrt{\log(1/\delta_\ell)}}$$

$\qquad\square$

**Definition A.3.** Define $\mathcal{E}_{var}$ as the event in which the following inequalities hold for all $\ell \in \mathbb{N}^+$ and $a \in \mathrm{Supp}(\pi_\ell)$ jointly:

$$\hat{\sigma}_\ell^2(a) \geq \frac{1}{4}\max\{\varepsilon_\ell, \sigma^2(a)\} \tag{1}$$

$$\hat{\sigma}_\ell^2(a) \leq \frac{3}{4}\sigma^2(a) + \frac{3}{4}\varepsilon_\ell \tag{2}$$

**Definition A.4.** Define $\mathcal{E}_{lss}$ as the event in which the following hold for all $\ell \geq 2$ and $a \in \mathcal{A}_\ell$ jointly:

$$|a^\top(\hat{\theta}^{(\ell)} - \theta^\star)| \leq \varepsilon_\ell$$

**Definition A.5.** Define $\mathcal{E}_{\mathcal{SR}steps}$ as the event in which the following holds for all $\ell \in \mathbb{N}^+$ and all $a \in \mathrm{Supp}(\pi_\ell)$ jointly:

$$T_\ell^{\mathcal{SR}}(a) \leq \frac{90}{\varepsilon_\ell}\log(2/\gamma_\ell)$$

**Definition A.6.** Define the good event $\mathcal{E}_{good}$ as follows: $\mathcal{E}_{var} \cap \mathcal{E}_{lss} \cap \mathcal{E}_{\mathcal{SR}steps}$.

**Lemma A.7.** *When running Algorithm 1 in phase $\ell$ of Algorithm 2, the following inequalities hold for all $a \in \mathrm{Supp}(\pi_\ell)$:*

$$\mathbb{E}[Z_s(a)] \geq \frac{1}{2} \max\{\varepsilon_\ell, \sigma^2(a)\} \tag{3}$$

$$\mathbb{E}[Z_s(a)] \leq \frac{1}{2}\sigma^2(a) + \frac{1}{2}\varepsilon_\ell \tag{4}$$

*Proof.* We first note that clearly $\mathbb{E}[Z_s(a)] \geq \varepsilon_\ell/2$. Also, since $\theta_s$ and $\theta_{s+1}$ are sampled i.i.d:

$$\mathbb{E}[Z_s(a)] \geq \mathbb{E}\left[\frac{1}{4}(a^\top \theta_s - a^\top \theta_{s+1})^2\right] = \frac{1}{4}\left(\mathbb{E}[(a^\top \theta_s)^2] - 2\mathbb{E}[a^\top \theta_s]\mathbb{E}[a^\top \theta_{s+1}] + \mathbb{E}[a^\top \theta_{s+1}]\right)$$

$$= \frac{1}{4}\left[\mathbb{E}[a^\top \theta_s]^2 - \mathbb{E}[a^\top \theta_s]^2\right] = \frac{1}{2}\mathrm{Var}(a^\top \theta_s) = \frac{1}{2}\sigma^2(a)$$

This concludes the proof of Equation (3). Similarly:

$$\mathbb{E}[Z_s(a)] \leq \frac{1}{4}\mathbb{E}[(a^\top \theta_s - a^\top \theta_{s+1})^2] + \frac{\varepsilon_\ell}{2} = \frac{1}{2}\sigma^2(a) + \frac{1}{2}\varepsilon_\ell$$

This concludes Equation (4) and the lemma. $\square$

We now restate and prove Lemma 3.5:

**Lemma A.8.** $\mathcal{E}_{var}$ *(Definition A.3) holds with probability $1 - \delta$.*

*Proof.* Fix some $\ell \in \mathbb{N}^+$ and $a \in \mathrm{Supp}(\pi_\ell)$. By the guarantees of the $\mathcal{SR}$ algorithm, with probability $1 - \gamma_\ell$:

$$|\hat{\sigma}_\ell^2(a) - \mathbb{E}[Z_s(a)]| \leq \frac{1}{2}\mathbb{E}[Z_s(a)] \tag{5}$$

Plugging Lemma A.7 into Equation (5) we get that:

$$\hat{\sigma}_\ell^2(a) \geq \frac{1}{2}\mathbb{E}[Z_s(a)] \geq \frac{1}{4}\max\{\varepsilon_\ell, \sigma^2(a)\}$$

And:

$$\hat{\sigma}_\ell^2(a) \leq \frac{3}{2}\mathbb{E}[Z_s(a)] \leq \frac{3}{2}\left(\frac{1}{2}\sigma^2(a) + \frac{1}{2}\varepsilon_\ell\right) = \frac{3}{4}\sigma^2(a) + \frac{3}{4}\varepsilon_\ell$$

By Kiefer & Wolfowitz (1960), we know that $|\mathrm{Supp}(\pi_\ell)| \leq d(d+1)/2$. Thus, using a union bound over $a \in \mathrm{Supp}(\pi_\ell)$ concludes our proof for $\ell$. A union bound over $\ell \in \mathbb{N}^+$ along with the fact that $\sum_{\ell=1}^{\infty} 1/\ell(\ell+1) = 1$ concludes our proof. $\square$

We continue by restating and proving Lemma 3.6:

**Lemma A.9.** *If $\mathcal{E}_{var}$ (Definition A.3) holds, then $\mathcal{E}_{lss}$ (Definition A.4) holds with probability $1 - \delta$.*

*Proof.* Fix some $\ell \in \mathbb{N}^+$ and $a \in \mathcal{A}_\ell$. Assume $\mathcal{E}_{var}$ holds. We first decompose $a^\top(\hat{\theta}^{(\ell)} - \theta^\star)$ to a sum of independent random variables. For all $s, \ell$ define:

$$Y_{s,\ell}(a) = a^\top V_\ell^{-1} \frac{1}{\hat{\sigma}_\ell^2(a_s)} a_s a_s^\top (\theta_s - \theta^\star)$$

And so:

$$\sum_{s=t_\ell}^{t_\ell + T_\ell} Y_{s,\ell}(a) = a^\top V_\ell^{-1}\left(\sum_{s=t_\ell}^{t_\ell + T_\ell} \frac{1}{\hat{\sigma}_\ell^2(a_s)} a_s a_s^\top \theta_s\right) - a^\top V_\ell^{-1}\left(\sum_{s=t_\ell}^{t_\ell + T_\ell} \frac{1}{\hat{\sigma}_\ell^2(a_s)} a_s a_s^\top\right)\theta^\star = a^\top(\hat{\theta} - \theta^\star)$$

Since $\mathbb{E}[\theta_s] = \theta^\star$, the random variables $Y_{s,\ell}(a)$ are zero-mean. We now proceed to bound their magnitude and variance. First, for the magnitude, it holds that:

$$|Y_{s,\ell}(a)| \le \frac{2}{\hat{\sigma}_\ell^2(a_s)} |a^\top V_\ell^{-1} a_s| \le \frac{2}{\hat{\sigma}_\ell^2(a_s)} \|a\|_{V_\ell^{-1}} \|a_s\|_{V_\ell^{-1}}$$
$$\le \frac{8\|a\|_{V_\ell^{-1}} \|a_s\|_{V_\ell^{-1}}}{\varepsilon_\ell} \le \frac{8\|a\|_{V_\ell^{-1}}}{7\sqrt{\log(1/\delta_\ell)}},$$

where the first inequality holds by our assumption that the rewards are bounded, the second by Hölder-Young, the third by $\mathcal{E}_{var}$ and the last by Lemma A.2.

Now, for the variance, again by $\mathcal{E}_{var}$:

$$\mathbb{E}[Y_{s,\ell}(a)^2] = \frac{1}{\hat{\sigma}_\ell^4(a_s)} a^\top V_\ell^{-1} a_s a_s^\top \Sigma a_s a_s^\top a = \frac{\sigma^2(a_s)}{\hat{\sigma}_\ell^4(a_s)} a^\top V_\ell^{-1} a_s a_s^\top a \le \frac{4}{\hat{\sigma}_\ell^2(a_s)} a^\top V_\ell^{-1} a_s a_s^\top V_\ell^{-1} a$$

And summing over all $s$ we get:

$$\sum_{s=t_\ell}^{t_\ell+T_\ell} \mathbb{E}[Y_{s,\ell}^2] \le 4a^\top V_\ell^{-1} \left( \sum_{s=t_\ell}^{t_\ell+T_\ell} \frac{1}{\hat{\sigma}_\ell^2(a_s)} a_s a_s^\top \right) V_\ell^{-1} a = 4a^\top V_\ell^{-1} a = 4\|a\|_{V_\ell^{-1}}^2$$

Now by Freedman (1975), with probability $1 - \delta_\ell$ it holds that:

$$a^\top(\hat{\theta}^{(\ell)} - \theta^\star) \le 4\sqrt{2}\|a\|_{V_\ell^{-1}} \sqrt{\log(1/\delta_\ell)} + \frac{2}{3} \cdot \frac{4\|a\|_{V_\ell^{-1}}}{7\sqrt{\log(1/\delta_\ell)}} \cdot \log(1/\delta_\ell)$$
$$\le 7\|a\|_{V_\ell^{-1}} \sqrt{\log(1/\delta_\ell)} \le \varepsilon_\ell,$$

where the second inequality follows from Lemma A.2. Similarly, one can show that this is also true for $a^\top(\theta^\star - \hat{\theta}^{(\ell)})$.

A union bound over all $a \in \mathcal{A}_\ell$ and $\ell \in \mathbb{N}^+$ yields our desired result. $\qquad \square$

**Lemma A.10.** *For all $\ell$ we have that:*

$$T_\ell \le \frac{d(d+1)}{2} + \frac{49d}{\varepsilon_\ell^2} \log(1/\delta_\ell) \hat{B}_\ell$$

*Proof.* By our choice of $T_\ell$ we have that:

$$T_\ell = \sum_{a \in \mathcal{A}_\ell} T_\ell(a) = \sum_{a \in \mathcal{A}_\ell} \left\lceil \frac{49d}{\varepsilon_\ell^2} \log(1/\delta_\ell) \hat{\sigma}_\ell^2(a) \pi_\ell(a) \right\rceil$$
$$\le |\text{Supp}(\pi_\ell)| + \frac{49d}{\varepsilon_\ell^2} \log(1/\delta_\ell) \hat{B}_\ell \le \frac{d(d+1)}{2} + \frac{49d}{\varepsilon_\ell^2} \log(1/\delta_\ell) \hat{B}_\ell,$$

where the last inequality follows by the bound on the support (Kiefer & Wolfowitz, 1960). $\qquad \square$

**Lemma A.11.** *Assume that $\mathcal{E}_{lss}$ holds. Let $a^\star \in \mathcal{A}$ be an optimal action. then for all $\ell \in \mathbb{N}^+$ jointly it holds that $a^\star \in \mathcal{A}_\ell$.*

*Proof.* By induction. The base case is trivial, since $\mathcal{A}_1 = \mathcal{A}$. We will show that the claim holds for $\ell + 1$. Denote $\hat{a}^{(\ell)} \in \arg\max_{a \in \mathcal{A}_\ell} a^\top \hat{\theta}^{(\ell)}$. Then:

$$(\hat{a}^{(\ell)} - a^\star)^\top \hat{\theta}^{(\ell)} \le (\hat{a}^{(\ell)} - a^\star)^\top (\hat{\theta}^{(\ell)} - \theta^\star) = (\hat{\theta}^{(\ell)} - \theta^\star)^\top \hat{a}^{(\ell)} - (a^\star)^\top (\hat{\theta}^{(\ell)} - \theta^\star) \le 2\varepsilon_\ell,$$

where the first inequality follows since $a^\star$ is optimal, and the second is since by our induction hypothesis, $a^\star \in \mathcal{A}_\ell$ and by $\mathcal{E}_{lss}$.

By our elimination criterion this concludes our proof. $\qquad \square$

**Lemma A.12.** *Assume that $\mathcal{E}_{lss}$ holds. Then, for all $a \in \mathcal{A}_\ell$ and $\ell \in \mathbb{N}^+$ jointly, we have that $\Delta_a \leq 8\varepsilon_\ell$.*

*Proof.* Assume $\mathcal{E}_{lss}$ holds and let $\ell \in \mathbb{N}^+$ and $a \in \mathcal{A}_\ell$. If $\ell = 1$, the claim holds trivially. Otherwise, let $\hat{a}^{(\ell-1)} \in arg\max_{b \in \mathcal{A}_{\ell-1}} b^\top \hat{\theta}^{(\ell-1)}$. We have that:

$$\Delta_a = (a^\star - a)^\top \theta^\star = (a^\star - a)^\top (\theta^\star - \hat{\theta}^{(\ell-1)}) + (a^\star - a)^\top \hat{\theta}^{(\ell-1)}$$
$$\leq a^{\star\top}(\theta^\star - \hat{\theta}^{(\ell-1)}) + \hat{\theta}^{(\ell-1)}(\hat{a}^{(\ell-1)} - a) + a^\top(\hat{\theta}^{(\ell-1)} - \theta^\star) \leq \varepsilon_{\ell-1} + 2\varepsilon_{\ell-1} + \varepsilon_{\ell-1} = 4\varepsilon_{\ell-1} = 8\varepsilon_\ell$$

Where the last inequality follows from $\mathcal{E}_{lss}$, the elimination criterion and Lemma A.11. $\qquad\square$

**Lemma A.13.** *Assume that $\mathcal{E}_{var}$ holds. Then we have that $M \leq \log(T)$.*

*Proof.* We have that:

$$T = \sum_{\ell=1}^{M} \left(T_\ell + T_\ell^{\mathcal{SR}}\right) \geq \sum_{\ell=1}^{M} T_\ell \geq \sum_{\ell=1}^{M} 49d \log(1/\delta_\ell) \frac{1}{\varepsilon_\ell^2} \sum_{a \in \mathcal{A}_\ell} \hat{\sigma}_\ell^2(a) \pi_\ell(a)$$
$$\geq \sum_{\ell=1}^{M} 12d \log(1/\delta_\ell) \frac{1}{\varepsilon_\ell^2} \varepsilon_\ell \sum_{a \in \mathcal{A}_\ell} \pi_\ell(a)$$
$$= 12d \log(1/\delta_\ell) \sum_{\ell=1}^{M} 2^\ell \geq 12d \log(1/\delta_M) 2^M,$$

where the third inequality is due to $\mathcal{E}_{var}$ and the last equality follows since $\pi_\ell$ is a design. Taking the base 2 logarithm and rearranging gives us:

$$M \leq \log\left(\frac{T}{12d \log(1/\delta_M)}\right) \leq \log(T).$$

$\qquad\square$

**Lemma A.14.** *$\mathcal{E}_{\mathcal{SR}steps}$ (Definition A.5) holds with probability at least $1 - \delta$.*

*Proof.* Fix some $\ell$ and $a \in \text{Supp}(\pi_\ell)$. By the guarantees of the $\mathcal{SR}$ algorithm, when running Algorithm 1 in phase $\ell$ of Algorithm 2, with probability $1 - \gamma_\ell$:

$$T_\ell^{\mathcal{SR}}(a) \leq \frac{45}{\mathbb{E}[Z_s(a)]} \log(2/\gamma_\ell) \tag{6}$$

By Lemma A.7 we have:

$$\mathbb{E}[Z_s(a)] \geq \frac{\varepsilon_\ell}{2}$$

Plugging this into Equation (6) we get that:

$$T_\ell^{\mathcal{SR}}(a) \leq \frac{90}{\varepsilon_\ell} \log(2/\gamma_\ell)$$

Using a union bound over all $\ell \in \mathbb{N}^+$ and $a \in \text{Supp}(\pi_\ell)$ concludes our proof. $\qquad\square$

**Lemma A.15.** *$\mathcal{E}_{good}$ holds with probability at least $1 - 3\delta$.*

*Proof.* Follows immediately from Lemma A.8, Lemma A.9 and Lemma A.14. $\qquad\square$

We are now ready for the final step of our proof, in which we bound the regret of Algorithm 2 and prove Theorem 3.1.

**Lemma A.16.** *With probability $1 - 3\delta$, the regret of Algorithm 2 is bounded by*

$$\mathcal{R}_T = \widetilde{O}\left(d^2 + d\log(K/\delta) + \sqrt{Td\log(K/\delta) \sum_{\ell=1}^{M} \text{tr}(\Sigma V(\pi_\ell))}\right)$$

*Proof.* Assume that $\mathcal{E}_{good}$ holds. Denote by $\ell_a$ the phase in which $a$ is eliminated. We have that:

$$
\begin{aligned}
\mathcal{R}_T &= \sum_{\ell=1}^{M} \sum_{a \in \mathcal{A}_\ell} T_\ell(a) \Delta_a + \sum_{\ell=1}^{M} \sum_{a \in \mathcal{A}_\ell} T_\ell^{\mathcal{SR}}(a) \Delta_a \\
&= \sum_{a \in \mathcal{A}} \sum_{\ell=1}^{\ell_a-1} T_\ell(a) \Delta_a + \sum_{a \in \mathcal{A}} \sum_{\ell=1}^{\ell_a-1} T_\ell^{\mathcal{SR}}(a) \Delta_a \\
&\le \sum_{a \in \mathcal{A}} \sum_{\ell=1}^{\ell_a-1} T_\ell(a) 8\varepsilon_\ell + \sum_{a \in \mathcal{A}} \sum_{\ell=1}^{\ell_a-1} T_\ell^{\mathcal{SR}}(a) 8\varepsilon_\ell \\
&\le 360 d(d+1) \log(2/\gamma_M) \log(T) + 8 \sum_{\ell=1}^{M} \varepsilon_\ell T_\ell,
\end{aligned}
\tag{7}
$$

where the first inequality is due to Lemma A.12 and the last inequality is by $\mathcal{E}_{\mathcal{SR}steps}$. We now proceed to bound the final term. Using Lemma A.10 and the fact that $\sum_{\ell=1}^{\infty} 1/2^i = 1$ we have:

$$
\begin{aligned}
\sum_{\ell=1}^{M} 8\varepsilon_\ell T_\ell &\le 4 \sum_{\ell=1}^{M} \varepsilon_\ell d(d+1) + 392 d \sum_{\ell=1}^{M} \log(1/\delta_\ell) \frac{1}{\varepsilon_\ell} \hat{B}_\ell \\
&\le 4d(d+1) + 392 d \sum_{\ell=1}^{M} \log(1/\delta_\ell) \frac{1}{\varepsilon_\ell} \hat{B}_\ell.
\end{aligned}
$$

Where $\hat{B}_\ell = \sum_{a \in \mathcal{A}_\ell} \hat{\sigma}_\ell^2(a) \pi_\ell(a)$. We continue to bound the final term:

$$
\begin{aligned}
392 d \sum_{\ell=1}^{M} \frac{\log(1/\delta_\ell)}{\varepsilon_\ell} \hat{B}_\ell &\le 392 d \sum_{\ell=1}^{M} \frac{\log(1/\delta_\ell)}{\varepsilon_\ell} \left( \sum_{a \in \mathcal{A}_\ell} \frac{3}{4} \sigma^2(a) \pi_\ell(a) + \frac{3}{4} \varepsilon_\ell \sum_{a \in \mathcal{A}_\ell} \pi_\ell(a) \right) \\
&\le 294 d \sum_{\ell=1}^{M} \frac{\log(1/\delta_\ell)}{\varepsilon_\ell} \sum_{a \in \mathcal{A}_\ell} \sigma^2(a) \pi_\ell(a) + 294 d \log(1/\delta_M) \log(T),
\end{aligned}
$$

where the first inequality is by $\mathcal{E}_{good}$, and the second is due to Lemma A.13.

Using the Cauchy-Schwarz inequality, we have that:

$$
\begin{aligned}
294 d \sum_{\ell=1}^{M} \frac{\log(1/\delta_\ell)}{\varepsilon_\ell} \sum_{a \in \mathcal{A}_\ell} \sigma^2(a) \pi_\ell(a) &= 294 \sum_{\ell=1}^{M} \varepsilon_\ell \frac{d \log(1/\delta_\ell)}{\varepsilon_\ell^2} B_\ell \\
&\le 294 \sqrt{\sum_{\ell=1}^{M} \varepsilon_\ell^2 d \log(1/\delta_\ell) \frac{1}{\varepsilon_\ell^2} B_\ell} \sqrt{\sum_{\ell=1}^{M} d \log(1/\delta_\ell) \frac{1}{\varepsilon_\ell^2} B_\ell} \\
&\le 294 \sqrt{\sum_{\ell=1}^{M} d \log(1/\delta_\ell) B_\ell} \cdot \sqrt{\frac{4}{49} T} \\
&= 84 \sqrt{d T \log(1/\delta_M) \sum_{\ell=1}^{M} \operatorname{tr}(\Sigma V(\pi_\ell))}
\end{aligned}
$$

Where the first equality is by the definition of $B_\ell$ and the first inequality follows from the Cauchy-Schwarz inequality. The second inequality is by our choice of $T_\ell$ and $\mathcal{E}_{var}$. The last equality is by Lemma A.1.

Plugging everything into Equation (7) and using Lemma A.1 we get that:

$$\mathcal{R}_T \leq 360d(d+1)\log(2/\gamma_M)\log(T) + 4d(d+1) + 294d\log(1/\delta_M)\log(T) + 84\sqrt{dT\log(1/\delta_M)\sum_{\ell=1}^{M}\text{tr}(\Sigma V(\pi_\ell))}$$

$$= \widetilde{O}\left(d^2 + d\log(K/\delta) + \sqrt{dT\log(K/\delta)\sum_{\ell=1}^{M}\text{tr}(\Sigma V(\pi_\ell))}\right)$$

The theorem follows since $\mathcal{E}_{good}$ holds with probability $1 - 3\delta$. $\qquad\square$

*Proof of Theorem 3.1.* In order to prove our theorem, we need to upper bound $\sum_{\ell=1}^{M}\text{tr}(\Sigma V(\pi_\ell))$. First, using Von Neumann's trace inequality (Mirsky, 1975), we have:

$$\text{tr}(\Sigma V(\pi_\ell)) \leq \text{tr}(\Sigma)\lambda_{max}(V(\pi_\ell)) \leq \text{tr}(\Sigma)\max_{a\in\mathcal{A}_\ell}\|a\|_2^2$$

Additionally, using the fact that $\sum_{a\in\mathcal{A}_\ell}\pi_\ell(a) = 1$, we have:

$$\text{tr}(\Sigma V(\pi_\ell)) = \sum_{a\in\mathcal{A}_\ell}\sigma^2(a)\pi_\ell(a) \leq \max_{a\in\mathcal{A}_\ell}\sigma^2(a)\sum_{a\in\mathcal{A}_\ell}\pi_\ell(a) = \max_{a\in\mathcal{A}_\ell}\sigma^2(a)$$

Thus, denoting $M_\sigma = \min\{\max_{a\in\mathcal{A}}\sigma^2(a), \ \max_{a\in\mathcal{A}}\|a\|_2^2\text{tr}(\Sigma)\}$, and combining the two bounds we have $\text{tr}(\Sigma V(\pi_\ell)) \leq M_\sigma$. Finally, plugging our bound into Lemma A.16 we have:

$$\mathcal{R}_T = \widetilde{O}\left(d^2 + d\log(K/\delta) + \sqrt{dT\log(K/\delta)M_\sigma}\right)$$

$\qquad\square$

# B. Regret Bound Proof for VALEE

In this section we prove the regret of Algorithm 3 as stated in Theorem 3.7 and Theorem 3.9.

The general flow of the proof for Theorem 3.9 is as follows:

1. Prove that $\hat{\sigma}_q^2$ is a good estimator for $\sigma_q^2$.

2. Prove that $\hat{N}$ is a good estimator of $\|\theta^\star\|_q$ and bound the number of steps we perform in the estimation.

3. Show that the $\hat{\theta}^{(j)}$ is a good estimator for $\theta^\star$.

4. Show that $\hat{a}$ is a good estimator for $a^\star$, i.e. has low sub-optimality gap.

5. Bound the regret of the algorithm.

Theorem 3.7 has the same proof structure, except for Item 1.

*Remark* B.1. Throughout our proof, we assume that $\|\theta^\star\|_q > 0$, since otherwise, any action achieves zero regret, and our regret bound holds trivially.

We start by proving some helpful lemmas.

**Lemma B.2.** $(q-1)p = q$

*Proof.* Since $\ell_p$ and $\ell_q$ are dual norms, we have that:

$$\frac{1}{q} + \frac{1}{p} = 1$$

Thus, by rearranging: $p + q = pq$, which means that: $q = p(q-1)$ $\qquad\square$

**Lemma B.3.** *For $q \geq 2$ we have that $\sigma_q^2 = \left( \sum_{i=1}^d \Sigma_{ii}^{q/2} \right)^{2/q} \leq 4$*

*Proof.* Denote $X_i = e_i^\top (\theta - \theta^\star)$ for $\theta$ sampled from $\nu$, and $X = (X_1, \ldots, X_d)$. We have that:

$$\Sigma_{ii} = \mathrm{Var}(X_i) = \mathbb{E}[X_i^2]$$

And so by Jensen:

$$\Sigma_{ii}^{q/2} = \mathbb{E}[X_i^2]^{q/2} \leq \mathbb{E}[|X_i|^q]$$

Summing over all $i \in [d]$ we have:

$$\sum_{i=1}^d \Sigma_{ii}^{q/2} \leq \mathbb{E}\Big[ \sum_{i=1}^d |X_i|^q \Big] = \mathbb{E}\Big[ \|X\|_q^q \Big].$$

Observing that: $X = \theta - \theta^\star$ we have that:

$$\sum_{i=1}^d \Sigma_{ii}^{q/2} \leq \mathbb{E}\Big[ \|\theta - \theta^\star\|_q^q \Big] \leq 2^{q-1} \mathbb{E}\Big[ \|\theta\|_q^q + \|\theta^\star\|_q^q \Big] \leq 2^q$$

Where the second inequality is since $|a - b| \leq 2^{q-1}(|a|^q + |b|^q)$ and the last is since $\nu$ is supported on the $\ell_q$ unit ball. Taking the $q$-th root concludes our proof. □

**Lemma B.4.** *For $q \geq 2$ and all $v = (v_1, \ldots, v_d) \in \mathbb{R}^d$, we have that:*

$$\frac{\|v\|_{q-1}}{\|v\|_q} \leq d^{1/q(q-1)}$$

*Proof.* We note that:

$$\|v\|_{q-1}^{q-1} = d \cdot \frac{1}{d} \sum_{i=1}^d \left( |v_i|^q \right)^{\frac{q-1}{q}} \leq d \Big( \frac{1}{d} \sum_{i=1}^d |v_i|^q \Big)^{\frac{q-1}{q}} = d \Big( \frac{1}{d} \|v\|_q^q \Big)^{\frac{q-1}{q}} = d^{1/q} \cdot \|v\|_q^{q-1},$$

where the inequality is by Jensen's inequality. Taking the $(q-1)$-th root and rearranging concludes our proof. □

**Lemma B.5.** *For all $\theta \in \mathbb{R}^d$ the action $a(\theta) \in \mathcal{A}$ such that:*

$$a(\theta)_i = \frac{sign(\theta_i)|\theta_i|^{q-1}}{\|\theta\|_q^{q-1}}$$

*is optimal with respect to $\theta$, i.e:*

$$a(\theta) \in \arg\max_{b \in \mathcal{A}} b^\top \theta.$$

*Proof.* By Lemma B.2, we have that:

$$\|a(\theta)\|_p^p = \sum_{i=1}^d \frac{sign(\theta_i)|\theta_i|^{(q-1)p}}{\|\theta\|_q^{(q-1)p}} = \sum_{i=1}^d \frac{sign(\theta_i)|\theta_i|^q}{\|\theta\|_q^q} = \frac{\|\theta\|_q^q}{\|\theta\|_q^q} = 1,$$

which means that $a(\theta)$ is feasible. We now show that it is optimal.

Since $\ell_p$ and $\ell_q$ are dual norms we have that:

$$\max_{b \in \mathcal{A}} b^\top \theta = \max_{b, \|b\|_p \leq 1} b^\top \theta = \|\theta\|_q.$$

Using this, we get:

$$a(\theta)^\top \theta = \sum_{i=1}^d \frac{sign(\theta_i)|\theta_i|^{q-1}}{\|\theta\|_q^{q-1}} \cdot \theta_i = \sum_{i=1}^d \frac{|\theta_i|^q}{\|\theta\|_q^{q-1}} = \frac{\|\theta\|_q^q}{\|\theta\|_q^{q-1}} = \|\theta\|_q = \max_{b \in \mathcal{A}} b^\top \theta.$$

Which concludes our proof. □

**Definition B.6.** Let $\mathcal{E}_{var}$ be the event in which the following hold jointly:

- The variance estimation is good: $\frac{1}{4}\sigma_q^2 \leq \hat{\sigma}_q^2 \leq \frac{3}{2}d^{2/q}\tau + \frac{3}{2}\sigma_q^2$.

- The regret of the estimation phase is bounded as: $\frac{180d}{\tau}\log(2d/\delta)$.

**Lemma B.7.** *For $q \geq 2$, event $\mathcal{E}_{var}$ (Definition B.6) holds with probability $1 - \delta$.*

*Proof.* Similarly to the proof of Lemma A.7 and Lemma A.8, we can show that with probability $1 - \delta$ the following holds for all $i \in [d]$ jointly:

$$\max\left\{\frac{\tau}{2}, \frac{\Sigma_{ii}}{4}\right\} \leq \hat{\sigma}^2(e_i) \leq \frac{3}{4}\Sigma_{ii} + \frac{3}{4}\tau.$$

Raising the inequality to the power of $q/2$ and using the fact that $(a + b)^{q/2} \leq 2^{q/2-1}(a^{q/2} + b^{q/2})$ we get:

$$\left(\frac{\Sigma_{ii}}{4}\right)^{q/2} \leq \left(\hat{\sigma}^2(e_i)\right)^{q/2} \leq 2^{q/2-1}\left(\frac{3}{4}\right)^{q/2}\left(\tau^{q/2} + \Sigma_{ii}^{q/2}\right).$$

Summing over all $i$ we have:

$$\frac{1}{4^{q/2}}\sum_{i=1}^{d}\Sigma_{ii}^{q/2} \leq \sum_{i=1}^{d}\left(\hat{\sigma}^2(e_i)\right)^{q/2} \leq d \cdot 2^{q/2-1}\left(\frac{3}{4}\right)^{q/2}\tau^{q/2} + 2^{q/2-1}\left(\frac{3}{4}\right)^{q/2}\sum_{i=1}^{d}\Sigma_{ii}^{q/2}.$$

Raising this to the power of $2/q$ and using the fact that $(a + b)^{2/q} \leq a^{2/q} + b^{2/q}$ we have:

$$\frac{1}{4}\sigma_q^2 \leq \hat{\sigma}_q^2 \leq \frac{3}{2}d^{2/q}\tau + \frac{3}{2}\sigma_q^2,$$

which proves the first part of the lemma. We now bound the regret of the variance estimation phase. Since $\mathbb{E}[Z_i] \geq \tau/2$, and again by the guarantees of the $\mathcal{SR}$ algorithm, with probability $1 - \delta/d$, the number of steps it performs for $e_i$ is bounded as:

$$T^{\mathcal{SR}}(e_i) \leq \frac{90}{\tau}\log(2d/\delta)$$

Using a union bound we get that this holds for all $i \in [d]$ jointly. Thus, the number of steps of the variance estimation step is bounded by $\frac{90d}{\tau}\log(2d/\delta)$, Since the instantaneous regret can be bounded by 2, this concludes our proof. □

*Remark* B.8. From this point on, we assume that $\mathcal{E}_{var}$ holds. Additionally, we assume that $\|\theta^\star\|_q > q\alpha^2\sigma_q^2$, since otherwise, given $\mathcal{E}_{var}$, any algorithm would yield:

$$\mathcal{R}_T \leq 2Tq\alpha^2\sigma_q^2 \leq 4\sqrt{dTq\kappa\sigma_q^2} = \widetilde{O}\left(\sqrt{dTq\log(1/\delta)\sigma_q^2}\right).$$

We now prove Lemma B.9, stated in Section 3.2.1, for the general $p \in (1, 2]$ and $1 \geq 2$ case.

**Lemma B.9.** *With probability $1 - \delta$, for all $i \in [d]$ and $j \leq \log(T)$ jointly, we have that:*

$$e_i^\top(\hat{\theta}^{(j)} - \theta^\star) \leq \sqrt{\Sigma_{ii}} \cdot \hat{\varepsilon}_j$$

*Proof.* Fix some $i \in [d]$ and a timestep $s$. Define the following random variables:

$$Y_{s,\ell,j,i} = \frac{1}{T_j^{exp}(e_i)}e_i^\top(\theta_s - \theta^\star)$$

We have that:

$$\sum_{s=t_{j,\ell}}^{t_{j,\ell}+T_j^{exp}(e_i)} Y_{s,\ell,j,i} = e_i^\top(\hat{\theta}^{(j,\ell)} - \theta^\star)$$

We will bound the variance of $Y_{s,\ell,j,i}$ and use the median of means estimator as seen in Lugosi & Mendelson (2019, Theorem 2). Since $Y_{s,\ell,j,i}$ are zero-mean, we have:

$$\mathrm{Var}(Y_{s,\ell,j,i}) = \frac{1}{T_j^{exp}(e_i)^2}\mathbb{E}[Y_{s,\ell,j,i}^2] = \frac{1}{T_j^{exp}(e_i)^2}e_i^\top \Sigma e_i = \frac{\Sigma_{ii}}{T_j^{exp}(e_i)^2}$$

Summing over all $s$:

$$\sum_{s=t_{j,\ell}}^{t_{j,\ell}+T_j^{exp}(e_i)}\mathbb{E}[Y_{s,\ell,j,i}^2] = \frac{\Sigma_{ii}}{T_j^{exp}(e_i)} = \frac{\hat{\varepsilon}_j^2 \Sigma_{ii}}{8}$$

Thus, by the Chebyshev inequality, with probability $3/4$:

$$e_i^\top(\hat{\theta}^{(j,\ell)} - \theta^\star) \leq \sqrt{\Sigma_{ii}}\cdot\hat{\varepsilon}_j \tag{8}$$

Now, let $Z_{i,j,\ell}$ be an indicator random variable that the inequality in Equation (8) doesn't hold. Denote the sum of these indicators as $Z_{i,j} = \sum_{\ell=1}^\kappa Z_{i,j,\ell}$. Then $Z_{i,j}$ is binomial with $\kappa$ trials and success probability at most $1/4$. Thus, $\mathbb{E}[Z_{i,j}] \leq \kappa/4$, and so using the standard median of means argument:

$$\mathbb{P}\left(e_i^\top(\hat{\theta}^{(j)} - \theta^\star) > \sqrt{\Sigma_{ii}}\cdot\hat{\varepsilon}_j\right) =$$
$$\mathbb{P}\left(\mathrm{Median}_{\ell\in[\kappa]}(e_i^\top\hat{\theta}^{(j,\ell)}) - \theta_i^\star > \sqrt{\Sigma_{ii}}\cdot\hat{\varepsilon}_j\right) = \mathbb{P}\left(\mathrm{Median}_{\ell\in[\kappa]}(e_i^\top(\hat{\theta}^{(j,\ell)} - \theta_i^\star)) > \sqrt{\Sigma_{ii}}\cdot\hat{\varepsilon}_j\right) \leq$$
$$\mathbb{P}\left(Z_{i,j} > \frac{k}{2}\right) \leq \mathbb{P}\left(Z_{i,j} - \mathbb{E}[Z_{i,j}] > \frac{k}{4}\right) \leq e^{-\kappa/8} = \frac{\delta}{d\log(T)}$$

Where the second equality is since the median is shift-invariant and the second inequality is by our bound on $\mathbb{E}[Z]$. The third inequality is by Hoeffding and the last equality is by our choice of $\kappa$.

A union bound over all $i \in [d]$ and $j \leq \log(T)$ concludes our proof. $\qquad\square$

We continue to prove Lemma 3.11, stated in Section 3.2.1, for the general $p \in (1,2]$ and $q \geq 2$ case:

**Lemma B.10.** *With probability $1-\delta$, for all $j \leq \log(T)$ jointly, we have that:*

$$\|\hat{\theta}^{(j)} - \theta^\star\|_q \leq \sqrt{\sigma_q^2}\hat{\varepsilon}_j$$

*Proof.* Let $x \in \mathbb{R}^d$ such that for $i \in [d]$:

$$x_i = \frac{\mathrm{sign}(\hat{\theta}_i^{(j)} - \theta_i^\star)|\hat{\theta}_i^{(j)} - \theta_i^\star|^{q-1}}{\|\hat{\theta}^{(j)} - \theta^\star\|_q^{q-1}}$$

We can see that:

$$\|x\|_p^p = \sum_{i=1}^d \frac{|\hat{\theta}_i^{(j)} - \theta_i^\star|^{(q-1)p}}{\|\hat{\theta}^{(j)} - \theta^\star\|_q^{(q-1)p}} = \frac{\|\hat{\theta}^{(j)} - \theta^\star\|_q^q}{\|\hat{\theta}^{(j)} - \theta^\star\|_q^q} = 1 \tag{9}$$

Where the second equality is by Lemma B.2. Since $\{e_i\}_{i\in[d]}$ is a basis, there exist some $\{\alpha_i\}_{i\in[d]} \subset \mathbb{R}$ such that $x = \sum_{i=1}^d \alpha_i e_i$. Thus, with probability $1-\delta$, for all $j \leq \log(T)$ jointly we have:

$$\|\hat{\theta}^{(j)} - \theta^\star\|_q = x^\top(\hat{\theta}^{(j)} - \theta^\star) = \sum_{i=1}^d \alpha_i e_i^\top(\hat{\theta}^{(j)} - \theta^\star) \leq \hat{\varepsilon}_j \sum_{i=1}^d |\alpha_i|\sqrt{\Sigma_{ii}}$$
$$\leq \hat{\varepsilon}_j\|x\|_p\sqrt{\sigma_q^2} = \sqrt{\sigma_q^2}\hat{\varepsilon}_j, \tag{10}$$

where the first inequality is by Lemma B.9, and the second is by Hölder-Young. The third equality is due to Equation (9).
$\qquad\square$

We continue to prove Lemma 3.14, stated in Section 3.2.1, for the general $p \in (1, 2]$ and $q \geq 2$ case:

**Lemma B.11.** *For $q \geq 2$, if the exploration stage of Algorithm 3 stops in iteration $L$, then with probability $1 - \delta$ we have:*

$$\frac{1}{4}\|\theta^\star\|_q \leq \hat{N}_L \leq \frac{5}{2}\|\theta^\star\|_q$$

*Proof.* By Lemma B.10, the reverse triangle inequality and the definition of $\hat{\varepsilon}_L$, with probability $1 - \delta$:

$$\left| \|\hat{\theta}^{(L)}\|_q - \|\theta^\star\|_q \right| \leq \sqrt{\sigma_q^2} \alpha \sqrt{\hat{N}_L}$$

By AM-GM:

$$\left| \|\hat{\theta}^{(L)}\|_q - \|\theta^\star\|_q \right| \leq \frac{1}{2}\left(\sigma_q^2 \alpha^2 + \hat{N}_L\right) \leq \frac{1}{2}\left(\frac{\|\theta^\star\|_q}{q} + \hat{N}_L\right) \leq \frac{1}{4}\|\theta^\star\|_q + \frac{1}{2}\hat{N}_L \tag{11}$$

Where the second inequality is by Remark B.8 and the last inequality is since $q \geq 2$.

Finally, by the stopping condition of the while loop we have that $\hat{N}_L \leq \|\hat{\theta}^{(L)}\|_q$. Thus, by rearranging, we get the upper bound:

$$\hat{N}_L \leq \frac{5}{2}\|\theta^\star\|_q.$$

For the lower bound, let us observe iteration $L - 1$, in which the while loop didn't stop. By the same argument as in Equation (11), we have:

$$\left| \|\hat{\theta}^{(L-1)}\|_q - \|\theta^\star\|_q \right| \leq \frac{1}{4}\|\theta^\star\|_q + \frac{1}{2}\hat{N}_{L-1}$$

And so by rearranging:

$$\|\theta^\star\|_q \leq \frac{4}{3}\|\hat{\theta}^{(L-1)}\|_q + \frac{2}{3}\hat{N}_{L-1}.$$

Since the algorithm didn't stop in this iteration, it holds that $\|\hat{\theta}^{(L-1)}\|_q < \hat{N}_{L-1}$. Using this, we get:

$$\|\theta^\star\|_q \leq 2\hat{N}_{L-1} = 4\hat{N}_L$$

$\square$

**Lemma B.12.** *For $q \geq 2$, with probability $1 - \delta$, the regret of the exploration stage of Algorithm 3 is bounded by:*

$$\widetilde{O}\left(\sqrt{dTq\log(1/\delta)\sigma_q^2} + d^{1/2+1/q}\sqrt{Tq\log(1/\delta)\cdot\tau}\right)$$

*Proof.* Let $L$ be the last iteration of the exploration stage. Applying the same argument as in Lemma B.11, we have:

$$\|\theta^\star\|_q \leq 4 \cdot 2^{-L}$$

Using this, denoting the regret of the exploration stage by $\mathcal{R}_T^{\text{explore}}$, with probability $1 - \delta$ we have:

$$\mathcal{R}_T^{\text{explore}} \leq d\log(T) + \sum_{j=1}^{L} \frac{8d\kappa}{\hat{\varepsilon}_j^2} \cdot 2\|\theta^\star\|_q = d\log(T) + \frac{16d\kappa\|\theta^\star\|_q}{\alpha^2} \sum_{j=1}^{L} \frac{1}{\hat{N}_j}$$

$$\leq d\log(T) + \frac{32d\kappa\|\theta^\star\|_q}{\alpha^2} \cdot 2^L \leq d\log(T) + \frac{32d\kappa}{\alpha^2} \cdot 4 \cdot 2^{-L} \cdot 2^L = d\log(T) + 128d\kappa\sqrt{\frac{Tq\hat{\sigma}_q^2}{d\kappa}}$$

$$= \widetilde{O}\left(d + \sqrt{dTq\log(1/\delta)\sigma_q^2} + d^{1/2+1/q}\sqrt{Tq\log(1/\delta)\tau}\right)$$

Where the last equality is by event $\mathcal{E}_{var}$ and Lemma B.7. $\square$

We continue to prove Lemma 3.12, stated in Section 3.2.1, for the general $p \in (1, 2]$ and $q \geq 2$ case:

**Lemma B.13.** *For $q \geq 2$, if Algorithm 3 reached the exploit stage after $L$ iterations, then with probability $1 - \delta$:*

$$\|\hat{a} - a^\star\|_p \leq \frac{3(q-1)\sqrt{\sigma_q^2 \hat{\varepsilon}_L}}{\|\theta^\star\|_q}$$

*Proof.* Throughout the proof, let $\phi(x) \in \mathbb{R}^d \mapsto \mathbb{R}^d$ be:

$$\phi(x)_i = \text{sign}(x_i)|x_i|^{q-1}$$

By Lemma B.5 and our choice of $\hat{a}$ we have that:

$$a^\star = \frac{\phi(\theta^\star)}{\|\theta^\star\|_q^{q-1}}, \quad \hat{a} = \frac{\phi(\hat{\theta}^{(L)})}{\|\hat{\theta}^{(L)}\|_q^{q-1}}$$

And so:

$$
\begin{aligned}
\|a^\star - \hat{a}\|_p &= \left\| \frac{\phi(\theta^\star)}{\|\theta^\star\|_q^{q-1}} - \frac{\phi(\hat{\theta}^{(L)})}{\|\theta^\star\|_q^{q-1}} + \frac{\phi(\hat{\theta}^{(L)})}{\|\theta^\star\|_q^{q-1}} - \frac{\phi(\hat{\theta}^{(L)})}{\|\hat{\theta}^{(L)}\|_q^{q-1}} \right\|_p \\
&\leq \left\| \frac{\phi(\theta^\star)}{\|\theta^\star\|_q^{q-1}} - \frac{\phi(\hat{\theta}^{(L)})}{\|\theta^\star\|_q^{q-1}} \right\|_p + \left\| \frac{\phi(\hat{\theta}^{(L)})}{\|\theta^\star\|_q^{q-1}} - \frac{\phi(\hat{\theta}^{(L)})}{\|\hat{\theta}^{(L)}\|_q^{q-1}} \right\|_p \\
&= \frac{\|\phi(\theta^\star) - \phi(\hat{\theta}^{(L)})\|_p}{\|\theta^\star\|_q^{q-1}} + \frac{\left| \|\hat{\theta}^{(L)}\|_q^{q-1} - \|\theta^\star\|_q^{q-1} \right|}{\|\theta^\star\|_q^{q-1}} \cdot \frac{\|\phi(\hat{\theta}^{(L)})\|_p}{\|\hat{\theta}^{(L)}\|_q^{q-1}} \\
&= \frac{\|\phi(\theta^\star) - \phi(\hat{\theta}^{(L)})\|_p}{\|\theta^\star\|_q^{q-1}} + \frac{\left| \|\hat{\theta}^{(L)}\|_q^{q-1} - \|\theta^\star\|_q^{q-1} \right|}{\|\theta^\star\|_q^{q-1}}
\end{aligned}
$$

Where the first inequality is by the triangle inequality, and the last equality is by Lemmas B.2 and B.5.

Using a symmetric argument, we can also show that:

$$\|a^\star - \hat{a}\|_p \leq \frac{\|\phi(\theta^\star) - \phi(\hat{\theta}^{(L)})\|_p}{\|\hat{\theta}^{(L)}\|_q^{q-1}} + \frac{\left| \|\hat{\theta}^{(L)}\|_q^{q-1} - \|\theta^\star\|_q^{q-1} \right|}{\|\hat{\theta}^{(L)}\|_q^{q-1}}$$

Combining the two bounds, we have:

$$\|a^\star - \hat{a}\|_p \leq \frac{\|\phi(\theta^\star) - \phi(\hat{\theta}^{(L)})\|_p}{\max\left\{ \|\hat{\theta}^{(L)}\|_q^{q-1}, \|\theta^\star\|_q^{q-1} \right\}} + \frac{\left| \|\hat{\theta}^{(L)}\|_q^{q-1} - \|\theta^\star\|_q^{q-1} \right|}{\max\left\{ \|\hat{\theta}^{(L)}\|_q^{q-1}, \|\theta^\star\|_q^{q-1} \right\}} \tag{12}$$

If $p = q = 2$, then $\|\phi(\theta^\star) - \phi(\hat{\theta}^{(L)})\|_p = \|\theta^\star - \hat{\theta}^{(L)}\|_q$, and so using the reverse triangle inequality and Lemma B.10 we get by Equation (12) that:

$$\|a^\star - \hat{a}\|_p \leq \frac{2\sqrt{\sigma_q^2 \hat{\varepsilon}_L}}{\|\theta^\star\|_q}$$

And we are done. From this point on, we assume that $q > 2$.

We proceed to bound each term of Equation (12) separately. By the mean value theorem, there exists some $z_i \in \left[ \min\{\hat{\theta}_i^{(L)}, \theta_i^\star\}, \max\{\hat{\theta}_i^{(L)}, \theta_i^\star\} \right]$ such that:

$$\phi(\theta_i^\star) - \phi(\hat{\theta}_i^{(L)}) = \phi'(z_i)(\theta_i^\star - \hat{\theta}_i^{(L)}) = (q-1)|z_i|^{q-2}(\theta_i^\star - \hat{\theta}_i^{(L)})$$

Taking the absolute value, raising both sides to the power of $p$ and summing over all $i \in [d]$ we get:

$$\|\phi(\theta^\star) - \phi(\hat{\theta}^{(L)})\|_p^p = (q-1)^p \sum_{i=1}^d |z_i|^{(q-2)p} |\theta_i^\star - \hat{\theta}_i^{(L)}|^p$$

Recalling that $q > 2$, by Hölder-Young using the dual norms $q - 1$ and $\frac{q-1}{q-2}$ we have:

$$\|\phi(\theta^\star) - \phi(\hat{\theta}^{(L)})\|_p^p \leq (q-1)^p \left( \sum_{i=1}^d |z_i|^{(q-1)p} \right)^{\frac{q-2}{q-1}} \left( \sum_{i=1}^d |\theta_i^\star - \hat{\theta}_i^{(L)}|^{(q-1)p} \right)^{\frac{1}{q-1}}$$

$$= (q-1)^p \left( \sum_{i=1}^d |z_i|^q \right)^{\frac{q-2}{q-1}} \left( \sum_{i=1}^d |\theta_i^\star - \hat{\theta}_i^{(L)}|^q \right)^{\frac{1}{q-1}}$$

Where the last equality is by Lemma B.2. Bounding $z_i$ we get that:

$$\|\phi(\theta^\star) - \phi(\hat{\theta}^{(L)})\|_p^p \leq (q-1)^p \left( \sum_{i=1}^d \max\left\{ |\hat{\theta}_i^{(L)}|, |\theta_i^\star| \right\}^q \right)^{\frac{q-2}{q-1}} \left\| \hat{\theta}^{(L)} - \theta^\star \right\|_q^{\frac{q}{q-1}}$$

$$\leq (q-1)^p \left( \sum_{i=1}^d |\hat{\theta}_i^{(L)}|^q + |\theta_i^\star|^q \right)^{\frac{q-2}{q-1}} \left\| \hat{\theta}^{(L)} - \theta^\star \right\|_q^{\frac{q}{q-1}}$$

$$= (q-1)^p \left( \|\hat{\theta}^{(L)}\|_q^q + \|\theta^\star\|_q^q \right)^{\frac{q-2}{q-1}} \left\| \hat{\theta}^{(L)} - \theta^\star \right\|_q^{\frac{q}{q-1}}$$

$$\leq (q-1)^p \cdot 2^{\frac{q-2}{q-1}} \cdot \max\left\{ \|\hat{\theta}^{(L)}\|_q^q, \|\theta^\star\|_q^q \right\}^{\frac{q-2}{q-1}} \cdot \left\| \hat{\theta}^{(L)} - \theta^\star \right\|_q^{\frac{q}{q-1}}$$

$$= (q-1)^p \cdot 2^{\frac{q-2}{q-1}} \cdot \max\left\{ \|\hat{\theta}^{(L)}\|_q^{\frac{q(q-2)}{q-1}}, \|\theta^\star\|_q^{\frac{q(q-2)}{q-1}} \right\} \cdot \left\| \hat{\theta}^{(L)} - \theta^\star \right\|_q^{\frac{q}{q-1}}$$

Where the second inequality is since $\max\{a, b\} \leq a + b$ and the third inequality is because $a + b \leq 2 \max\{a, b\}$.

Taking the $p$-th root and using Lemma B.2 again we have:

$$\|\phi(\theta^\star) - \phi(\hat{\theta}^{(L)})\|_p \leq (q-1) 2^{\frac{q-2}{q}} \cdot \max\left\{ \|\hat{\theta}^{(L)}\|_q^{q-2}, \|\theta^\star\|_q^{q-2} \right\} \cdot \|\hat{\theta}^{(L)} - \theta^\star\|_q$$

$$\leq (q-1) 2 \max\left\{ \|\hat{\theta}^{(L)}\|_q^{q-2}, \|\theta^\star\|_q^{q-2} \right\} \cdot \|\hat{\theta}^{(L)} - \theta^\star\|_q$$

For the second term of Equation (12), again by the mean value theorem with $f(x) = x^{q-1}$, there exists some $z \in \left[ \min\{\|\theta^\star\|_q, \|\hat{\theta}^{(L)}\|_q\}, \max\{\|\theta^\star\|_q, \|\hat{\theta}^{(L)}\|_q\} \right]$ such that:

$$\left| \|\hat{\theta}^{(L)}\|_q^{q-1} - \|\theta^\star\|_q^{q-1} \right| = (q-1) z^{q-2} \left| \|\hat{\theta}^{(L)}\|_q - \|\theta^\star\|_q \right| \leq (q-1) \left( \max\left\{ \|\theta^\star\|_q, \|\hat{\theta}^{(L)}\|_q \right\} \right)^{q-2} \left| \|\hat{\theta}^{(L)}\|_q - \|\theta^\star\|_q \right|$$

And by the reverse triangle inequality we have:

$$\left| \|\hat{\theta}^{(L)}\|_q^{q-1} - \|\theta^\star\|_q^{q-1} \right| \leq (q-1) \left( \max\left\{ \|\theta^\star\|_q, \|\hat{\theta}^{(L)}\|_q \right\} \right)^{q-2} \|\hat{\theta}^{(L)} - \theta^\star\|_q$$

$$= (q-1) \max\left\{ \|\theta^\star\|_q^{q-2}, \|\hat{\theta}^{(L)}\|_q^{q-2} \right\} \|\hat{\theta}^{(L)} - \theta^\star\|_q$$

Plugging both bounds into Equation (12) and using Lemma B.10 we get that:

$$\|a^\star - \hat{a}\|_p \leq \frac{3(q-1) \max\left\{ \|\theta^\star\|_q, \|\hat{\theta}^{(L)}\|_q \right\}^{q-2}}{\max\left\{ \|\theta^\star\|_q, \|\hat{\theta}^{(L)}\|_q \right\}^{q-1}} \cdot \sqrt{\sigma_q^2 \hat{\varepsilon}_L} = \frac{3(q-1) \sqrt{\sigma_q^2} \hat{\varepsilon}_L}{\|\theta^\star\|_q}$$

Which concludes our proof. □

We continue prove Lemma 3.13, stated in Section 3.2.1, for the general $p \in (1, 2]$ and $q \geq 2$ case:

**Lemma B.14.** *For $q \geq 2$, if Algorithm 3 reached the exploit stage after $L$ iterations, then with probability $1 - \delta$, the suboptimality gap is bounded by:*

$$\Delta_{\hat{a}} \leq \frac{3(q-1)\sigma_q^2 \hat{\varepsilon}_L^2}{\|\theta^\star\|_q}$$

*Proof.* It holds that:

$$\Delta_{\hat{a}} = (a^\star - \hat{a})^\top \theta^\star \leq (a^\star - \hat{a})^\top (\theta^\star - \hat{\theta}^{(L)}) \leq \|a^\star - \hat{a}\|_p \|\theta^\star - \hat{\theta}^{(L)}\|_q$$

Where the first inequality is since $\hat{a}$ is optimal with respect to $\hat{\theta}$ according to Lemma B.5 and the second is by Hölder-Young. Now, by Lemma B.10 and Lemma B.13, we have with probability $1 - \delta$:

$$\Delta_{\hat{a}} \leq \frac{3(q-1)\sqrt{\sigma_q^2 \hat{\varepsilon}_L}}{\|\theta^\star\|_q} \cdot \sqrt{\sigma_q^2 \hat{\varepsilon}_L}$$

Which concludes our proof. $\square$

**Lemma B.15.** *For $q \geq 2$, with probability $1 - \delta$, the regret of the exploitation stage is bounded by:*

$$\widetilde{O}\left(\sqrt{dTq \log(1/\delta)\sigma_q^2}\right)$$

*Proof.* Denote the regret of the exploitation stage by $\mathcal{R}_T^{\text{exploit}}$. By Lemma B.14, with probability $1 - \delta$:

$$\mathcal{R}_T^{\text{exploit}} \leq T \cdot \Delta_{\hat{a}} \leq T \cdot \frac{3(q-1)\sigma_q^2 \hat{\varepsilon}_L^2}{\|\theta^\star\|_q}$$

Thus, with probability $1 - \delta$:

$$\mathcal{R}_T^{\text{exploit}} \leq \frac{3T(q-1)}{\|\theta^\star\|_q} \cdot \sigma_q^2 \alpha^2 \hat{N}_L \leq 8T(q-1)\alpha^2 \sigma_q^2$$

$$\leq 8\sqrt{dTq \log(d\log(T)/\delta)} \cdot \frac{\sigma_q^2}{\sqrt{\hat{\sigma}_q^2}} = \widetilde{O}\left(\sqrt{dTq \log(1/\delta)\sigma_q^2}\right)$$

Where the second inequality is by Lemma B.11 and the last equality is by $\mathcal{E}_{var}$. $\square$

We are now ready to prove the main result of this section, which is Theorem 3.9.

*Proof of Theorem 3.9.* Denote the regret of the variance estimation stage by $\mathcal{R}_T^{\text{var-est}}$. By Lemma B.7, Lemma B.12 and Lemma B.15 we get that with probability $1 - \delta$:

$$\mathcal{R}_T = \mathcal{R}_T^{\text{var-est}} + \mathcal{R}_T^{\text{explore}} + \mathcal{R}_T^{\text{exploit}}$$
$$= \widetilde{O}\left(\frac{45d\log(2d/\delta)}{\tau}\right) + \widetilde{O}\left(d + \sqrt{dTq\log(1/\delta)\sigma_q^2} + d^{1/2+1/q}\sqrt{Tq\log(1/\delta) \cdot \tau}\right) + \widetilde{O}\left(\sqrt{dTq\log(1/\delta)\sigma_q^2}\right)$$

Optimizing for $\tau$ (disregarding constants), we get that:

$$\tau = \frac{d^{1/3-2/3q}\log^{1/3}(d/\delta)}{T^{1/3}q^{1/3}}$$

Substituting this for $\tau$ in our regret bound, we have:

$$\mathcal{R}_T = \widetilde{O}\left(d + d^{2/3+2/3q}\log^{2/3}(1/\delta)q^{1/3}T^{1/3} + \sqrt{dTq\log(1/\delta)\sigma_q^2}\right)$$

Which concludes our proof. $\square$

For the case of a known covariance matrix, we derive a corollary, stated as Theorem 3.7.

*Proof of Theorem 3.7.* The analysis in the case of a known covariance matrix is the same as in Theorem 3.9, except that we don't need to estimate the variance. Thus, removing the regret of the variance estimation phase and the regret originating from the variance estimation error, it can be shown that with probability $1 - \delta$ we have:

$$\mathcal{R}_T = \widetilde{O}\Big(d + \sqrt{dTq \log(1/\delta)\sigma_q^2}\Big)$$

which concludes our proof. □

## C. Lower Bounds

In this section, we prove lower bounds on the stochastic model with parameter noise, as stated in Theorem 4.1, Theorem 4.3 and Theorem 4.4. We follow a similar construction as in (Shamir, 2014) and (Bubeck et al., 2017), where they prove a lower bound for adversarial linear bandits. Throughout this section, we denote by $r_t$ the reward at time $t$.

We start by proving Theorem 4.1.

*Proof of Theorem 4.1.* We follow the construction and proof methodology of (Shamir, 2014). We Define a distribution $\mathcal{D}_\xi$, where $\xi$ is chosen uniformly at random from $\{-1, 1\}^d$ and $\mathcal{D}_\xi \sim \mathcal{N}(\varepsilon\xi, \frac{\sigma^2}{d^{2/q}} \cdot \mathbb{I}_d)$, where $\varepsilon = d^{1/2-1/q}\sigma/\sqrt{T}$. It is sufficient to prove the claim any deterministic algorithm and the result follows for a randomized algorithm.

First, we show that $\mathcal{D}_\xi$ is valid. We clearly have that:

$$\Big(\sum_{i=1}^d \Sigma_{ii}^{q/2}\Big)^{2/q} = \Big(\sum_{i=1}^d \Big(\frac{\sigma^2}{d^{2/q}}\Big)^{q/2}\Big)^{2/q} = \sigma^2$$

Also:

$$\theta_\xi^\star = \mathbb{E}[\mathcal{D}_\xi] = \varepsilon\xi$$

And so:

$$\|\theta_\xi^\star\|_q = \|\varepsilon\xi\|_q = \varepsilon\|\xi\|_q = \varepsilon d^{1/q} = \frac{d^{1/2}\sigma}{\sqrt{T}} \leq 1$$

Where the last inequality is by our choice of $\varepsilon$, since $T \geq d$. We now prove our regret lower bound. By Lemma B.5 the optimal action with respect to $\mathcal{D}_\xi$ is given by $a_\xi^\star = d^{(1-q)/q}\xi$. To avoid clutter, from this point on we refer to $\theta_\xi^\star$ and $a_\xi^\star$ as $\theta^\star$ and $a^\star$ respectively.

Denote by $\mathcal{T}_0$ the timesteps $t$ in which $a_t = 0$, and let $\mathcal{T} = [T] \setminus \mathcal{T}_0$. For $t \in \mathcal{T}_0$, the instantaneous regret is $\|\theta^\star\|_q = \varepsilon d^{1/q}$, and so we have that:

$$\mathbb{E}[\mathcal{R}_T(\mathcal{ALG}, \mathcal{D}_\xi)] = |\mathcal{T}_0|\varepsilon d^{1/q} + \sum_{t \in \mathcal{T}} \mathbb{E}\Big[(a^\star - a_t)^\top \theta^\star\Big] \tag{13}$$

We now proceed to bound the second term, which we denote by $\mathcal{R}_T(\mathcal{ALG}, \mathcal{D}_\xi, \mathcal{T})$. Let $\bar{a} = \frac{1}{T}\sum_{t \in \mathcal{T}} a_t$. Since $\theta^\star = \varepsilon d^{(q-1)/q}a^\star$, we have:

$$\mathcal{R}_T(\mathcal{ALG}, \mathcal{D}_\xi, \mathcal{T}) = |\mathcal{T}| \cdot a^{\star\top}\theta^\star - \mathbb{E}\Big[\sum_{t \in \mathcal{T}} a_t^\top \theta^\star\Big] = |\mathcal{T}|\mathbb{E}\Big[\varepsilon d^{(q-1)/q}\|a^\star\|_2^2 - \varepsilon d^{(q-1)/q}\bar{a}^\top a^\star\Big]$$

$$= |\mathcal{T}|\varepsilon d^{(q-1)/q}\mathbb{E}\Big[\|a^\star\|_2^2 - \bar{a}^\top a^\star\Big]$$

We now bound the expectation. Plugging in the value of $a^\star$ we have:

$$\mathbb{E}\Big[\|a^\star\|_2^2 - \bar{a}^\top a^\star\Big] = \mathbb{E}\Big[d^{\frac{2-q}{q}} - \bar{a}^\top \xi d^{\frac{1-q}{q}}\Big] = \mathbb{E}\Big[d^{\frac{2-q}{q}} - d^{\frac{1-q}{q}}\sum_{i=1}^d \bar{a}_i\xi_i\Big] = \sum_{i=1}^d \mathbb{E}\Big[d^{\frac{2-2q}{q}} - d^{\frac{1-q}{q}}\bar{a}_i\xi_i\Big]$$

$$= \underbrace{\sum_{i=1}^d \mathbb{E}\Big[d^{\frac{2-2q}{q}} - d^{\frac{1-q}{q}}\bar{a}_i\xi_i \mid \bar{a}_i\xi_i > 0\Big]\mathbb{P}(\bar{a}_i\xi_i > 0)}_{E_1} + \underbrace{\sum_{i=1}^d \mathbb{E}\Big[d^{\frac{2-2q}{q}} - d^{\frac{1-q}{q}}\bar{a}_i\xi_i \mid \bar{a}_i\xi_i \leq 0\Big]\mathbb{P}(\bar{a}_i\xi_i \leq 0)}_{E_2}$$

Where the last equality is by the law of total expectation. Now, we claim that $E_1 \geq 0$. It suffices to show that:

$$\sum_{i=1}^{d} \mathbb{E}[\bar{a}_i \xi_i \mid \bar{a}_i \xi_i > 0] \mathbb{P}(\bar{a}_i \xi_i > 0) \leq d^{1/q}$$

Indeed, we have:

$$\sum_{i=1}^{d} \mathbb{E}[\bar{a}_i \xi_i \mid \bar{a}_i \xi_i > 0] \mathbb{P}(\bar{a}_i \xi_i > 0) \leq \sum_{i=1}^{d} \mathbb{E}[\bar{a}_i \xi_i \mid \bar{a}_i \xi_i > 0] = \mathbb{E}[\bar{a}^\top \xi] \leq \mathbb{E}\left[\|\bar{a}\|_p \|\xi\|_q\right] \leq d^{1/q}$$

Where the second inequality is by Hölder-Young and the last is since $\|a_t\|_p \leq 1$. Using the fact that $E_1 \geq 0$, we have:

$$\mathcal{R}_T(\mathcal{ALG}, \mathcal{D}_\xi, \mathcal{T}) \geq |\mathcal{T}| \varepsilon d^{\frac{q-1}{q}} E_2 \geq |\mathcal{T}| \varepsilon d^{\frac{1-q}{q}} \sum_{i=1}^{d} \mathbb{P}(\bar{a}_i \xi_i \leq 0)$$

Now fix $i \in [d]$. Observing the probability distribution, we have:

$$\mathbb{P}(\hat{a}_i \xi_i \leq 0) = \frac{1}{2}\Big(\mathbb{P}(\bar{a}_i \leq 0 \mid \xi_i = 1) + \mathbb{P}(\bar{a}_i \geq 0 \mid \xi_i = -1)\Big) = \frac{1}{2}\Big(\mathbb{P}(\bar{a}_i \leq 0 \mid \xi_i = 1) + \mathbb{P}(\bar{a}_i \geq 0 \mid \xi_i = -1)\Big)$$

$$= \frac{1}{2}\Big(1 - \Big(\mathbb{P}(\bar{a}_i \leq 0 \mid \xi_i = -1) - \mathbb{P}(\bar{a}_i \leq 0 \mid \xi_i = 1)\Big)\Big)$$

$$\geq \frac{1}{2}\Big(1 - \Big|\mathbb{P}(\bar{a}_i \leq 0 \mid \xi_i = -1) - \mathbb{P}(\bar{a}_i \leq 0 \mid \xi_i = 1)\Big|\Big)$$

Plugging this into our lower bound we have:

$$\mathcal{R}_T(\mathcal{ALG}, \mathcal{D}_\xi, \mathcal{T}) \geq \frac{|\mathcal{T}| \varepsilon d^{1/q}}{2}\Big(1 - \sum_{i=1}^{d} \frac{1}{d}\Big|\mathbb{P}(\bar{a}_i \leq 0 \mid \xi_i = -1) - \mathbb{P}(\bar{a}_i \leq 0 \mid \xi_i = 1)\Big|\Big) \tag{14}$$

Using our assumption that $\mathcal{ALG}$ is deterministic, the action $\bar{a}$ is deterministically determined given the rewards $r = (r_1, \ldots, r_T)$. Thus, denoting $\bar{a} = \bar{a}(r)$ and $f$ as the PDF of the distribution over rewards $(r_1, \ldots, r_T)$, we can lower bound:

$$\mathcal{R}_T(\mathcal{ALG}, \mathcal{D}_\xi, \mathcal{T}) \geq \frac{|\mathcal{T}| \varepsilon d^{1/q}}{2}\Big(1 - \sum_{i=1}^{d} \frac{1}{d}\Big|\int_r \mathbb{1}_{\{\bar{a}_i(r) \leq 0\}} \Big(f(r \mid \xi_i = -1) - f(r \mid \xi_i = 1)\Big)\Big|\Big)$$

$$\geq \frac{|\mathcal{T}| \varepsilon d^{1/q}}{2}\Big(1 - \sum_{i=1}^{d} \frac{1}{d}\int_r \mathbb{1}_{\{\bar{a}_i(r) \leq 0\}}\Big|f(r \mid \xi_i = -1) - f(r \mid \xi_i = 1)\Big|\Big) \tag{15}$$

Denoting the integral as $\mathcal{I}$, we observe that $\mathcal{I}$ is the total variation distance between the distribution over reward vectors given $\xi_i = 1$ and given $\xi_i = -1$. Thus, by Pinsker's inequality:

$$\mathcal{I} \leq \sqrt{2KL\Big(f(r \mid \xi_i = -1)\Big|\Big|f(r \mid \xi_i = 1)\Big)}$$

And by the chain rule of $KL$ divergence this is equal to:

$$\sqrt{2\sum_{t=1}^{T} KL\Big(f(r_t \mid \xi_i = -1, r_{1:t-1})\Big|\Big|f(r_t \mid \xi_i = 1, r_{1:t-1})\Big)} \tag{16}$$

We proceed to calculate the $KL$ divergence of each summand. Clearly, for $t \in \mathcal{T}_0$, the KL divergence is 0, and so Equation (16) is equal to:

$$\sqrt{2\sum_{t \in \mathcal{T}} KL\Big(f(r_t \mid \xi_i = -1, r_{1:t-1})\Big|\Big|f(r_t \mid \xi_i = 1, r_{1:t-1})\Big)}$$

By our choice of $\mathcal{D}_\xi$, for all $t \in \mathcal{T}$ we have that:

$$a_t^\top \theta_t | \xi_i = -1, r_{1:t-1} \sim \mathcal{N}\left(\varepsilon \sum_{j=1, j \neq i}^{d} \xi_j a_{t,j} - \varepsilon a_{t,i}, \frac{\sigma^2}{d^{2/q}} \sum_{j=1}^{d} a_{t,j}^2\right)$$

And similarly:

$$a_t^\top \theta_t | \xi_i = 1, r_{1:t-1} \sim \mathcal{N}\left(\varepsilon \sum_{j=1, j \neq i}^{d} \xi_j a_{t,j} + \varepsilon a_{t,i}, \frac{\sigma^2}{d^{2/q}} \sum_{j=1}^{d} a_{t,j}^2\right)$$

Thus, it can be verified that the $KL$ divergence is as follows:

$$KL\left(f(r_t \mid \xi_i = -1, r_{1:t-1}) \middle\| f(r_t \mid \xi_i = 1, r_{1:t-1})\right) = \mathbb{E}\left[\frac{2d^{2/q}\varepsilon^2 a_{t,i}^2}{\sigma^2 \sum_{j=1}^{d} a_{t,j}^2} \middle| \xi_i = -1\right]$$

And so:

$$\mathcal{I} \leq \sqrt{\sum_{t \in \mathcal{T}} \mathbb{E}\left[\frac{2d^{2/q}\varepsilon^2 a_{t,i}^2}{\sigma^2 \sum_{j=1}^{d} a_{t,j}^2} \middle| \xi_i = -1\right]}$$

By replacing the roles of the two distributions we can get a symmetric bound, which implies that:

$$\mathcal{I} \leq \min\left\{\sqrt{\sum_{t=1}^{T} \mathbb{E}\left[\frac{2d^{2/q}\varepsilon^2 a_{t,i}^2}{\sigma^2 \sum_{j=1}^{d} a_{t,j}^2} \middle| \xi_i = -1\right]}, \sqrt{\sum_{t=1}^{T} \mathbb{E}\left[\frac{2d^{2/q}\varepsilon^2 a_{t,i}^2}{\sigma^2 \sum_{j=1}^{d} a_{t,j}^2} \middle| \xi_i = 1\right]}\right\}$$

$$\leq \sqrt{\sum_{t=1}^{T} \mathbb{E}\left[\frac{d^{2/q}\varepsilon^2 a_{t,i}^2}{\sigma^2 \sum_{j=1}^{d} a_{t,j}^2} \middle| \xi_i = -1\right] + \sum_{t=1}^{T} \mathbb{E}\left[\frac{d^{2/q}\varepsilon^2 a_{t,i}^2}{\sigma^2 \sum_{j=1}^{d} a_{t,j}^2} \middle| \xi_i = 1\right]}$$

$$= \frac{\sqrt{2}\varepsilon d^{1/q}}{\sigma}\sqrt{\sum_{t=1}^{T} \frac{1}{2}\left(\mathbb{E}\left[\frac{a_{t,i}^2}{\sum_{j=1}^{d} a_{t,j}^2} \middle| \xi_i = -1\right] + \mathbb{E}\left[\frac{a_{t,i}^2}{\sum_{j=1}^{d} a_{t,j}^2} \middle| \xi_i = 1\right]\right)}$$

$$= \frac{\sqrt{2}\varepsilon d^{1/q}}{\sigma}\sqrt{\sum_{t=1}^{T} \mathbb{E}\left[\frac{a_{t_i}^2}{\sum_{j=1}^{d} a_{t,j}^2}\right]}$$

Where the inequality follows since $\min\{\sqrt{a}, \sqrt{b}\} \leq \sqrt{a + b}/\sqrt{2}$ and the last equality follows since $\xi_i$ is uniform over $\{-1, 1\}$.

Plugging this into Equation (15) and using Jensen's inequality we get:

$$\mathcal{R}_T(\mathcal{ALG}, \mathcal{D}_\xi, \mathcal{T}) \geq \frac{|\mathcal{T}|\varepsilon d^{1/q}}{2}\left(1 - \frac{\sqrt{2}\varepsilon d^{1/q}}{\sigma}\sqrt{\frac{1}{d}\sum_{t \in \mathcal{T}} \mathbb{E}\left[\frac{\sum_{i=1}^{d} a_{t_i}^2}{\sum_{j=1}^{d} a_{t,j}^2}\right]}\right) = \frac{|\mathcal{T}|\varepsilon d^{1/q}}{2}\left(1 - \frac{\sqrt{2}\varepsilon d^{1/q-1/2}\sqrt{|\mathcal{T}|}}{\sigma}\right)$$

$$\geq \frac{|\mathcal{T}|\varepsilon d^{1/q}}{2}\left(1 - \frac{\sqrt{2}\varepsilon d^{1/q-1/2}\sqrt{T}}{\sigma}\right)$$

Where the last inequality is since $|\mathcal{T}| \leq T$. Setting $\varepsilon = d^{1/2-1/q}\sigma/2\sqrt{2T}$ yields:

$$\mathcal{R}_T(\mathcal{ALG}, \mathcal{D}_\xi, \mathcal{T}) \geq \frac{\sigma\sqrt{d}|\mathcal{T}|}{20\sqrt{T}}$$

Plugging this into Equation (13) we have:

$$\mathbb{E}[\mathcal{R}_T(\mathcal{ALG}, \mathcal{D}_\xi)] \geq \frac{\sigma\sqrt{d}|\mathcal{T}_0|}{2\sqrt{2T}} + \frac{\sigma\sqrt{d}|\mathcal{T}|}{20\sqrt{T}}$$

Noticing that it must be that $|\mathcal{T}_0| = \Omega(T)$ or $|\mathcal{T}| = \Omega(T)$, we have that:

$$\mathbb{E}[\mathcal{R}_T(\mathcal{ALG}, \mathcal{D}_\xi)] = \Omega(\sqrt{dT\sigma^2})$$

Finally, using Shamir, 2014, Theorem 8, we can make sure that our distribution is supported on the unit ball almost surely, with a cost of $\sqrt{\log(T)}$. This implies a lower bound of:

$$\Omega\left(\frac{\sqrt{dT\sigma^2}}{\sqrt{\log(T)}}\right)$$

which concludes our proof. $\qquad\square$

We continue to prove Theorem 4.3.

*Proof of Theorem 4.3.* We follow the construction and proof methodology of (Bubeck et al., 2017). For convenience and ease of notation we will work over $\mathbb{R}^{d+1}$. We Define a distribution $\mathcal{D}_\xi$, where $\xi$ is chosen uniformly at random from $\{-1,1\}^d$. For $\theta = (w, z) \sim \mathcal{D}_\xi$ we define $w \sim \mathcal{N}(1, \sigma^2/2)$ and $z \sim \mathcal{N}(\varepsilon\xi, \frac{\sigma^2}{2d^{2/q}} \cdot \mathbb{I}_d)$, where $\varepsilon^q = \sigma/C\sqrt{T}$ with $C$ to be set later. It is sufficient to prove the claim for any deterministic algorithm and the result follows for a randomized algorithm.

First, we show that $\mathcal{D}_\xi$ is valid, up to scaling. Clearly $\sigma_q^2 = \sigma^2$. Also: $\theta_\xi^\star = (w^\star, z^\star) = \mathbb{E}[\mathcal{D}_\xi] = (1, \varepsilon\xi)$ And so:

$$\|\theta_\xi^\star\|_q = \|(1, \varepsilon\xi)\|_q = 1 + d\varepsilon^q \leq 1 + d^{1/q}\varepsilon \leq 2$$

Where the last inequality is by our assumption on $T$.

We now prove our regret lower bound. Denote $a_t = (x_t, y_t)$, where $x_t \in \mathbb{R}$ and $y_t \in \mathbb{R}^d$. Denote $(\bar{x}, \bar{y}) = \sum_{t=1}^T \mathbb{E}[(x_t, y_t)]$. Also let $a^\star = (x^\star, y^\star) = arg\max_{a\in\mathcal{A}} a^\top\theta^\star$.

We start by restating two key lemmas from Bubeck et al. (2017). We say a coordinate is wrong if $\bar{y}_i\xi_i \leq 0$.

**Lemma C.1** (restatement of Lemma 6 in Bubeck et al. (2017)). *Let $s$ be the number of wrong coordinates. Then:*

$$\mathbb{E}[\mathcal{R}_T] \geq \frac{1}{4}\varepsilon^q sT$$

**Lemma C.2** (restatement of Lemma 7 in Bubeck et al. (2017)). *If $\bar{x} \leq 1 - 4\varepsilon^q d$, then:*

$$\mathbb{E}[\mathcal{R}_T] \geq \varepsilon^q dT$$

We observe that $r_t|r_{1:t-1} \sim \mathcal{N}(x_t + \varepsilon\xi y_t, \sigma_t^2)$, where:

$$\sigma_t^2 = \frac{\sigma^2 x_t^2}{2} + \frac{\sigma^2}{2d^{2/q}}\|y_t\|_2^2$$

And so, using Pinsker's inequality, and calculating the KL between two Gaussians, with a similar argument to Theorem 4.1, we have:

$$\mathcal{TV}\left(f(r_t \mid \xi_i = -1, r_{1:t-1})\middle\|f(r_t \mid \xi_i = 1, r_{1:t-1})\right) \leq \sqrt{\sum_{t=1}^T \mathbb{E}\left[\frac{2\varepsilon^2 y_{t,i}^2}{\sigma_t^2}\right]}$$

Where $\mathcal{TV}$ is the total variation distance.

Using the standard coin tossing total variation error lower bound and Jensen's inequality, we get:

$$\mathbb{E}\left[\frac{1}{T}\sum_{t=1}^T\sum_{i=1}^d \mathbb{1}_{\{y_{t,i}\xi_i\leq 0\}}\right] \geq \frac{d}{2} - \sqrt{d\sum_{t=1}^T\mathbb{E}\left[\frac{2\varepsilon^2\|y_t\|_2^2}{\sigma_t^2}\right]} \tag{17}$$

We note that the LHS is the average (over time) number of coordinates for which the learner chose a wrong sign, and by Lemma C.1 we know this controls the regret. Thus, similarly to Bubeck et al. (2017), it is sufficient to prove that:

$$\sum_{t=1}^{T} \mathbb{E}\left[\frac{2\varepsilon^2\|y_t\|_2^2}{\sigma_t^2}\right] \leq d/5$$

Since this would imply that Equation (17) is $\Omega(d)$. We proceed to bound the LHS, by dividing the timesteps to "exploration" and "exploitation" round. Define $t$ as an exploration round if $x_t < 1/4$, and an exploitation round otherwise. Denote the set of exploration rounds by $E_1$ and the set of exploitation rounds by $E_2$. We have that:

$$\sum_{t=1}^{T} \mathbb{E}\left[\frac{2\varepsilon^2\|y_t\|_2^2}{\sigma_t^2}\right] \leq \underbrace{\sum_{t\in E_1} \mathbb{E}\left[\frac{2\varepsilon^2\|y_t\|_2^2}{\sigma_t^2}\right]}_{S_1} + \underbrace{\sum_{t\in E_2} \mathbb{E}\left[\frac{2\varepsilon^2\|y_t\|_2^2}{\sigma_t^2}\right]}_{S_2}$$

We proceed to bound each term individually. For $t \in E_1$, we have:

$$\sigma_t^2 \geq \frac{\sigma^2\|y_t\|_2^2}{2d^{2/q}}$$

And so:

$$S_1 \leq \frac{4\varepsilon^2 d^{2/q}}{\sigma^2} \sum_{t\in E_1} 1 = \frac{4\varepsilon^2 d^{2/q}}{\sigma^2} \sum_{t=1}^{T} \mathbb{1}_{\{x_t < 1/4\}}$$

We note that:

$$\mathcal{R}_T = \Omega\left(\sum_{t=1}^{T} \mathbb{1}_{\{x_t < 1/4\}}\right)$$

and consider two cases. If $\sum_{t=1}^{T} \mathbb{1}_{\{x_t < 1/4\}} \geq d\sqrt{T\sigma^2}$, then we are done. Otherwise:

$$\frac{4\varepsilon^2 d^{2/q}}{\sigma^2} \sum_{t=1}^{T} \mathbb{1}_{\{x_t < 1/4\}} \leq \frac{4\varepsilon^2 d^{1+2/q}\sqrt{T}}{\sigma} \leq \frac{4}{C}\sigma^{2/q-1} d^{1+2/q} T^{1/2-1/q} \leq \frac{4}{C}d$$

Where the last inequality holds for $T \geq d^{\frac{4}{2-q}}$ and since $\sigma^{2/q-1} \leq 1$.

For $t \in E_2$, we have:

$$\sigma_t^2 \geq \frac{\sigma^2 x_t^2}{2} \geq \frac{1}{32}\sigma^2$$

Now, by Hölder's inequality and since $\|a_t\|_p \leq 1$:

$$\|y_t\|_2^2 \leq d^{1-2/p}\|y_t \leq d^{1-2/p}\|_p(1-|x_t|^p)^{2/p}$$

And so:

$$S_2 \leq \frac{64\varepsilon^2 \sum_{t\in E_2}}{\sigma^2} \mathbb{E}\left[\|y_t\|_2^2\right] \leq \frac{64d^{1-2/p}\varepsilon^2}{\sigma^2} \sum_{t\in E_2} (1-|x_t|^p)^{2/p}$$

Now, by the mean-value theorem, there exists $z \in [0,1]$ such that:

$$1 - x_t^p \leq pz^{p-1}(1-x_t) \leq p(1-x_t)$$

Plugging this into our previous bound and using Jensen, we get:

$$S_2 \leq \frac{64d^{1-2/p}\varepsilon^2}{\sigma^2}p^{2/p} \sum_{t\in E_2} (1-|x_t|)^{2/p} \leq \frac{64d^{1-2/p}\varepsilon^2}{\sigma^2}p^{2/p} \sum_{t=1}^{T} (1-|x_t|)^{2/p} \leq \frac{64d^{1-2/p}\varepsilon^2}{\sigma^2}p^{2/p}T(1-\bar{x})^{2/p}$$

By Lemma C.2 we can assume that $\bar{x} \geq 1 - 4\varepsilon^q d$. Using this and since $p^{2/p} \leq 3$, we have;

$$S_2 \leq \frac{64d^{1-2/p}\varepsilon^2}{\sigma^2}p^{2/p}T(4\varepsilon^q d)^{2/p} \leq \frac{768dT\varepsilon^{2+2q/p}}{\sigma^2} = \frac{768dT\varepsilon^{2q}}{\sigma^2} = \frac{768}{C^2}d$$

Choosing $C \geq 73$ concludes our upper bound for $S_2$ and the proof of the theorem. $\qquad\square$

Finally, we prove Theorem 4.4.

*Proof of Theorem 4.4.* The same proof as Theorem 4.3 applies, noticing that the maximal variance of $D_\xi$ is attained with action $e_1$, for which $\sigma^2(e_1) = \sigma^2/2$. $\qquad\square$

