# OpenReview forum: "Stochastic Linear Bandits with Parameter Noise"
_ICML.cc/2026/Conference — ICML 2026 regular_

### Official Review · Reviewer_U5NX · 2026-02-27

**Soundness:** 3
**Presentation:** 4
**Significance:** 3
**Originality:** 3
**Overall Recommendation:** 5
**Confidence:** 4

**Summary:**

The paper studies a variant of adversarial and stochastic linear bandits by considering a linear reward vector that is random at each time instant and sampled from an unknown measure (although the second moment is assumed known in some applications). The authors provide an algorithm for  discrete arm sets and also for more general continuous domains that contain a standard hypercube. Lower bounds are also provided, although only in the case that $p\in (1,2]$ and with the second moment assumed known.

**Compliance With Llm Reviewing Policy:**

Affirmed.

**Final Justification:**

I remain positive about the paper and vote for its acceptance.

**Key Questions For Authors:**

1. An interesting question is how this setting relates to Bayesian linear bandits, where $\theta$ is assumed to be random but fixed throughout the algorithm. Another difference in the Bayesian setting is that there is still additive noise, so I guess it's not clear to me that the Bayesian setting would imply this one, or vice versa. I think a brief discussion in the related work section would be nice.

2. Although I see that the worst-case scenario of variant-dependent methods would perform worse than the authors algorithm, I do wonder to what extent those methods are adaptive. More specifically, could they zoom into directions aligned with the smaller eigenvectors of $\mathbb{E}[\theta\theta^{\top}]$ while keeping the regret performance? Intuitively this seems likely. Average eigenvalue of $\mathbb{E}[\theta\theta^{\top}]$ is O(1/d) so basically if the actions are "averagely exploratory" this should give  $\mathcal{O}(\sqrt{dT})$, which would be the same rate as the authors claim. In any case I don't think this would be applicable to the continous domain.

**Limitations:**

Yes

**Strengths And Weaknesses:**

Presentation: I found the paper quite easy and interesting to read. Allowing the reward vector to vary i.i.d. over time instances has never occurred to me as a way to bring a significant "structural" change to the usual badnit problem.  Both the introduction and the related work section serve well to interest the reader into why the problem might be relevant. The variance dependent section in related works section has brought to my attention by this work so that was especially appricated. The algorithms and design decisions are also clearly stated, for example, why a simple discretization argument on $\ell_2$ ball would be insufficient to obtain order-optimal accuracy. The limitations are also stated in the open, knowledge of the second moment is not obfuscated or hidden.

Significance: Although the paper is very much a theoretical contribution with no immediate applications, I still find the studied problem to be significant with a few immediate (non-trivial) follow-ups.  How far can we push the bound with unknown covariance? What can be done when $\theta$ is in the Hilbert space? The question is also mathematically interesting and the new setting did ask for some novelty in algortihm design.

Originality: As stated above, the idea of the bandit framework where $\theta$ varies i.i.d temporally is a novel and non-trivial extension of existing frameworks. Actions here can also play a role in variance reduction, introducing another trade-off between the mean and variance of the reward. The idea of using OLS weighted by estimated arm variances in a discrete arm setting is quite cool and, to my knowledge, novel. For the continuous algorithm variant, although timing exploration based on the norm of the theta estimate is quite standard, using the dual norm here is also interesting and new, in my opinion.

Soundness: The paper is technically sound, the algorithm design somewhat follows the usual structure in bandit literature and the ideas behind the proofs are clearly stated and intuitively "fit" with the design of the algorithm.

---

> ### Author Rebuttal · Authors · 2026-03-29
>
> * **W1**: Significance: Although the paper is very much a theoretical contribution with no immediate applications, I still find the studied problem to be significant with a few immediate (non-trivial) follow-ups. How far can we push the bound with unknown covariance? What can be done when $\theta$ is in the Hilbert space? The question is also mathematically interesting and the new setting did ask for some novelty in algortihm design.
>
>   **A1.1**: The unknown covariance gap is actually relevant for a very specific case, in which the variance is extremely small as a function of T. When the variance is constant or just larger than $T^{-1/3}$, the additive term becomes a low order term, and our result is optimal. Hence, in the interesting and realistic case, where the variances are lower bounded by a constant, our bound becomes tight. On the other extreme, when the variance is zero (deterministic rewards), the problem becomes very easy. The challenge, which we were not able to fully address, is when the variance is extremely small as a function of T, but not zero. In this case, we suffer a small additive penalty of $T^{1/3}$, which is still better than the non variance dependent bound of $\sqrt{dT}$.
>
>   **A1.2**: Even if the rewards are deterministic, the regret is lower bounded by d, so we believe that we can’t do much in general Hilbert spaces.
>
> * **Q1**: An interesting question is how this setting relates to Bayesian linear bandits, where  is assumed to be random but fixed throughout the algorithm. Another difference in the Bayesian setting is that there is still additive noise, so I guess it's not clear to me that the Bayesian setting would imply this one, or vice versa. I think a brief discussion in the related work section would be nice.
>
>   **A**: Thank you for the remark, we will add such a discussion in the final version.
>
> * **Q2**: Although I see that the worst-case scenario of variant-dependent methods would perform worse than the authors algorithm, I do wonder to what extent those methods are adaptive. More specifically, could they zoom into directions aligned with the smaller eigenvectors of $E[\theta \theta^\top]$ while keeping the regret performance? Intuitively this seems likely. Average eigenvalue of $E[\theta \theta^\top]$ is O(1/d) so basically if the actions are "averagely exploratory" this should give $O(\sqrt{dT})$, which would be the same rate as the authors claim. In any case I don't think this would be applicable to the continous domain.
>
>   **A**: This sounds like an interesting future direction, and we would be glad if you could elaborate more.
>
>   To the best of our knowledge, existing algorithms are not proved to adapt to the small variance in the parameter noise model. We suspect this is not the case and finding a concrete counter example might be interesting.

---

> > ### Author Rebuttal · Reviewer_U5NX · 2026-04-01
> >
> > Regarding A1.2: The effective rank of the covariance operator is usually bounded far below $T$ and, in many applications, plays the role of the "effective dimension". It would be interesting to see if $d$ here could be replaced by the effective dimension, and what type of algorithms would be necessary.

---

### Official Review · Reviewer_9QGv · 2026-03-08

**Soundness:** 4
**Presentation:** 2
**Significance:** 3
**Originality:** 4
**Overall Recommendation:** 5
**Confidence:** 3

**Summary:**

The paper develops regret bounds for stochastic linear bandits problem with parameter noise model with different types of actions sets (with associated Algorithms VASE and VALEE). The paper further shows that the bound for $l_p$ unit-ball based action set is optimal up to logarithmic factors, and is attainable using an explore-exploit algorithm (VALEE).

**Compliance With Llm Reviewing Policy:**

Affirmed.

**Final Justification:**

I preserve may positive evaluation and the score.

**Key Questions For Authors:**

1. When the covariance matrix is unknown, an extra factor of $dT^{1/3}$ appears. It is unclear to me whether this is an artifact of the algorithm or an inherent limitation of the model itself.
2. The assumption of $|a^\top \theta| < 1$ could be discussed more elaborately along with a discussion of real-time systems which satisfy this.
3. Can the authors elaborate on a specific use-case which can be reasonably modeled using an additive parameter noise framework?
4. Can the authors elaborate on the choice of $\tau$ in Algorithm 1 and its effect on the properties demonstrated by the algorithm?

**Limitations:**

Yes.

**Strengths And Weaknesses:**

Strengths:
1. The paper is theoretically very rigorous in its treatment of the results.
1.  The matching upper and lower bounds (upto log factors) for the $l_p$ based norm balls, and that it can be achieved using a reasonably simple algorithm is an interesting result.

Weaknesses:
1. Thought the results are well presented, the flow of the paper is at times, inconsistent, making it hard to follow the text (for instance, many notations seem undefined, some are defined after they are used).

---

> ### Author Rebuttal · Authors · 2026-03-29
>
> * **W1**: Thought the results are well presented, the flow of the paper is at times, inconsistent, making it hard to follow the text (for instance, many notations seem undefined, some are defined after they are used).
>
>   **A**: Thank you for the remark, we will improve the presentation in the final version.
>
> * **Q1**: When the covariance matrix is unknown, an extra factor of $dT^{1/3}$ appears. It is unclear to me whether this is an artifact of the algorithm or an inherent limitation of the model itself.
>
>   **A**: We believe that the additive factor in the case of an unknown covariance matrix is an artifact of the simple explore-exploit algorithm, and not an inherent limitation of the model. As mentioned in the future work section, “we believe this can’t be mitigated with an explore-exploit algorithm, finding an algorithm which is optimal when the covariance matrix is unknown is left to future work.”
>
> * **Q2**: The assumption of $|a^\top \theta| < 1$ could be discussed more elaborately along with a discussion of real-time systems which satisfy this.
>
>   **A**: It is clear that we need to make some assumptions about the boundedness of the realized rewards. Our assumption is standard in linear bandits and was also used in the bandit algorithms books. Alternatively, we can assume that the reward is bounded by some constant M, and the results follow similarly (with M in the regret).
>
> * **Q3**: Can the authors elaborate on a specific use-case which can be reasonably modeled using an additive parameter noise framework?
>
>   **A**: We believe that the parameter noise model is natural and can model numerous use cases. One example we mention in the introduction is online advertising. If we consider $\theta_t$ as a feature vector that represents a user sampled from the population, and our goal is to find an advertising strategy that works well in expectation across all users.
>
> * **Q4**: Can the authors elaborate on the choice of $\tau$ in Algorithm 1 and its effect on the properties demonstrated by the algorithm?
>
>   **A**: In order to bound the regret and runtime of Algorithm 1, we need to lower bound the variance of the reward of the actions. For this purpose, we introduced $\tau$, to get an upper bound of $1/\tau$ on the number of the iterations of Algorithm 1.

---

> > ### Author Rebuttal · Reviewer_9QGv · 2026-04-02
> >
> > Thank you for the discussion.

---

### Official Review · Reviewer_8rZg · 2026-03-10

**Soundness:** 3
**Presentation:** 3
**Significance:** 3
**Originality:** 3
**Overall Recommendation:** 4
**Confidence:** 1

**Summary:**

This paper studies stochastic linear bandits under parameter noise. The paper argues this model is structurally different from additive-noise linear bandits because the realized reward remains linear and the learner can affect reward variance through the chosen action. This paper provides variance-aware successive-elimination algorithm for finite action sets and simple variance-aware explore-exploit algorithm for $\ell_p$ ball. It obtains matching or near-matching lower bounds showing minimax-optimal variance-dependent regret in several regimes.

**Compliance With Llm Reviewing Policy:**

Affirmed.

**Key Questions For Authors:**

1. Can you clarify exactly which lower bound supports the finite-action optimality claim in the abstract/introduction? As written, I could not map that claim cleanly to a theorem statement.
2. Could we use FTRL or OMD to obtain the best of both results?

**Strengths And Weaknesses:**

Strengths:

1. The parameter-noise model is less studied than additive-noise or adversarial linear bandits, and the paper makes a convincing case that it is both natural and technically distinct.
2. The discussion contrasting additive noise, parameter noise, and adversarial settings is one of the strongest parts of the paper. The explanation of why parameter noise can be easier than additive noise, and why variance-aware exploration helps, is clear and conceptually useful.
3. Even without checking every appendix detail, the high-level proof ideas are plausible and well motivated.


Weaknesses:

1. Unknown-covariance result is not fully closed. For $p \leq 2$, the unknown- $\Sigma$ case still has an additive $d^{2 / 3+2 / 3 q} T^{1 / 3}$-type term, and the paper explicitly leaves closing this gap as future work. That is a real limitation relative to the cleaner known-covariance result.

---

> ### Author Rebuttal · Authors · 2026-03-29
>
> * **W1**: Unknown-covariance result is not fully closed. For $p \leq 2$, the unknown- $\Sigma$ case still has an additive $d^{2/3+2/3q} T^{1/3}$-type term, and the paper explicitly leaves closing this gap as future work. That is a real limitation relative to the cleaner known-covariance result.
>
>   **A**: The unknown covariance gap is actually relevant for a very specific case, in which the variance is extremely small as a function of T. When the variance is constant or just larger than $T^{-1/3}$, the additive term becomes a low order term, and our result is optimal. Hence, in the interesting and realistic case, where the variances are lower bounded by a constant, our bound becomes tight. On the other extreme, when the variance is zero (deterministic rewards), the problem becomes very easy. The challenge, which we were not able to fully address, is when the variance is extremely small as a function of T, but not zero. In this case, we suffer a small additive penalty of $T^{1/3}$, which is still better than the non variance dependent bound of $\sqrt{dT}$.
>
> * **Q1**: Can you clarify exactly which lower bound supports the finite-action optimality claim in the abstract/introduction? As written, I could not map that claim cleanly to a theorem statement.
>
>   **A**: Theorem 4.4 shows that there exists an action set, i.e. an $\ell_p$ unit ball for $p > 2$, for which the regret is lower bounded by $d\sqrt{T \sigma_{max}^2}$. Theorem 3.1 shows that VASE has a regret bound of $\sqrt{dT \sigma_{max}^2 log(K)}$ for a finite action set of size K. By a covering argument on the continuous $\ell_p$ unit ball with $\log(K) \approx d$, VASE gives us a regret bound of $d\sqrt{T\sigma_{max}^2}$, which matches the lower bound.
>
> * **Q2**: Could we use FTRL or OMD to obtain the best of both results?
>
>   **A**: FTRL and OMD are designed for adversarial environments and can’t be used for achieving variance aware bounds in a stochastic environment.

---

> > ### Author Rebuttal · Reviewer_8rZg · 2026-04-05
> >
> > Thank you for the discussion. I keep my score.

---

### Official Review · Reviewer_3yVr · 2026-03-13

**Soundness:** 3
**Presentation:** 3
**Significance:** 2
**Originality:** 3
**Overall Recommendation:** 4
**Confidence:** 3

**Summary:**

This paper studies stochastic linear bandits under a parameter-noise model, where the reward is generated by a random linear parameter rather than by standard additive noise. The paper proposes a variance-aware design/elimination algorithm (VASE) for finite action sets, and a simple coordinate-wise explore-then-commit algorithm (VALEE) for $\ell_p$ balls. The main results are variance-dependent regret bounds, including a near-tight characterization for $\ell_p$ balls with $p \le 2$, together with corresponding lower bounds. The paper is technically solid and clearly written.

**Compliance With Llm Reviewing Policy:**

Affirmed.

**Final Justification:**

My concerns have been adequately addressed. I have increased my score.

**Key Questions For Authors:**

- What is the main conceptual takeaway beyond reproducing the known $\ell_p$ phase transition in a different stochastic model? Should the contribution mainly be viewed as a model-separation result between additive noise and parameter noise?
- Is there a more general geometric or statistical condition underlying the success of VALEE for $p \le 2$? At present, the result feels quite specific to $\ell_p$ balls and coordinate-wise probing.
- Can the authors better clarify the independent value of the variance-aware G-optimal design component, e.g., whether it is expected to be useful beyond the present parameter-noise setting?

**Limitations:**

yes

**Strengths And Weaknesses:**

Strengths
- The paper gives a clean variance-dependent analysis in the parameter-noise model, and the $\ell_p$ results for $p \le 2$ are fairly sharp, especially with the accompanying lower bounds.
- The variance-aware G-optimal design component is reasonably interesting: estimating variances on the design support and using them for both sample allocation and weighted regression is a clean technical combination.

Weaknesses
- The main conceptual phenomenon seems closely tied to prior work on adversarial linear bandits over $\ell_p$ balls, especially the $p \le 2$ versus $p > 2$ phase transition. As a result, the paper reads more as a variance-dependent refinement in a different stochastic model than as a fundamentally new explanation.
- The algorithmic novelty appears somewhat limited. VASE mainly combines familiar ingredients (G-optimal design, elimination, variance estimation, weighted least squares), while VALEE is a very simple coordinate-wise explore-then-commit procedure.
- The paper does not seem to extract a more general or transferable principle beyond this specific model. In particular, it remains unclear what should be expected for broader action geometries or more general noise structures.

---

> ### Author Rebuttal · Authors · 2026-03-29
>
> * **W2**: The algorithmic novelty appears somewhat limited. VASE mainly combines familiar ingredients (G-optimal design, elimination, variance estimation, weighted least squares), while VALEE is a very simple coordinate-wise explore-then-commit procedure.
>
>   **A**: The algorithmic novelty in VASE is leveraging variance estimation for tuning the number of times each arm is selected. The algorithmic novelty in VALEE is that a very simple algorithm is optimal.This is in contrast to the adversarial model,in which explore-exploit always fails.
>
> * **Q1+W1**: What is the main conceptual takeaway beyond reproducing the known  phase transition in a different stochastic model? Should the contribution mainly be viewed as a model-separation result between additive noise and parameter noise?
>
>   **A**: We show that while the additive and parameter noise models seem similar, they behave differently, have different regret bounds and have different algorithms that are optimal. While the phase transition is similar to that in the adversarial model, the additive noise model surprisingly doesn’t have a phase transition, and has $d \sqrt{T}$ regret for all $\ell_p$ balls, which further distinguishes the two models.
>
> * **Q2+W3**: Is there a more general geometric or statistical condition underlying the success of VALEE for $p \leq 2$? At present, the result feels quite specific to $\ell_p$ balls and coordinate-wise probing.
>
>   **A**: Generally, we believe that VALEE works for any gauge set, which is a strongly convex set that contains 0 . For a gauge set with a known parameter J, VALEE works trivially. The challenge is gauge sets with an unknown parameter. $\ell_p$ unit balls with $p \leq 2$ are specific examples of such sets, for which the parameter is the $||\theta^*||_q$, where q is the dual norm. In this case, we were able to prove that VALEE works, without knowledge of the parameter. We believe that generalizing this to any gauge set is not trivial, and we were not able to address this due to time constraints.
>
> * **Q3**: Can the authors better clarify the independent value of the variance-aware G-optimal design component, e.g., whether it is expected to be useful beyond the present parameter-noise setting?
>
>   **A**: We introduced the variance aware G-optimal design to get a non-algorithm dependent regret bound in the parameter noise setting. This can potentially be used in other settings where G-optimal design is used, to get refined variance dependent bounds.

---

> > ### Author Rebuttal · Reviewer_3yVr · 2026-04-03
> >
> > I thank the authors for their rebuttal. My concerns have been adequately addressed. I have increased my score.

---

### Decision · Program_Chairs · 2026-04-30

**Decision:**

Accept (regular)

**Comment:**

The paper analyzes the stochastic linear bandit problem where the additive noise is absent but instead, the parameter is stochastic with fixed mean and covariance. In this setting, different arms have different variance. The paper derives variance-dependent regret lower bounds for (i) the general, fixed action set case and (ii) the case where the action set is the $\ell_p$-unit ball with $p\in(1,2]$, and proposes two algorithms whose regret upper bounds match the lower bounds for (i) and (ii), respectively, even without the knowledge of the variance. Both algorithms integrate the stopping rule algorithm to efficiently estimate the unknown variances. The first algorithm extends the successive elimination algorithm of Lattimore and Szepesvari (2020) to the case where arms have heterogeneous and unknown variance, while the second algorithm employs a simple explore-exploit scheme with a doubling trick to adapt to the norm of the mean parameter and Median-Of-Means estimator to efficiently estimate the mean parameter.

The reviewers agree that the paper studies an underexplored parameter-noise problem and clearly distinguishes it from additive-noise and adversarial bandit settings, with a compelling motivation for its importance. They appreciate that the paper provides lower bounds and proposes algorithms achieving sharp regret bounds relative to these lower bounds. The authors are encouraged to further improve the clarity and flow of the presentation.